# Altered structural brain asymmetry in autism spectrum disorder in a study of 54 datasets

Merel C. Postema [ID] et al.[#]

Altered structural brain asymmetry in autism spectrum disorder (ASD) has been reported. However, findings have been inconsistent, likely due to limited sample sizes. Here we investigated 1,774 individuals with ASD and 1,809 controls, from 54 independent data sets of the ENIGMA consortium. ASD was significantly associated with alterations of cortical thickness asymmetry in mostly medial frontal, orbitofrontal, cingulate and inferior temporal areas, and also with asymmetry of orbitofrontal surface area. These differences generally involved reduced asymmetry in individuals with ASD compared to controls. Furthermore, putamen volume asymmetry was significantly increased in ASD. The largest case-control effect size was Cohen's $d = -0.13$, for asymmetry of superior frontal cortical thickness. Most effects did not depend on age, sex, IQ, severity or medication use. Altered lateralized neurodevelopment may therefore be a feature of ASD, affecting widespread brain regions with diverse functions. Large-scale analysis was necessary to quantify subtle alterations of brain structural asymmetry in ASD.

[#]A full list of authors and their affiliations appears at the end of the paper.

Autism spectrum disorder (ASD) is an umbrella diagnosis, capturing several previously separate pervasive developmental disorders with various levels of symptom severity, including Autistic Disorder, Asperger's Syndrome, Childhood Disintegrative Disorder, and Pervasive Developmental Disorder—Not Otherwise Specified (PDD-NOS)[1]. According to the Diagnostic and Statistical Manual of Mental Disorders (DSM) version 5, diagnosis of ASD requires the presence of at least three symptoms of impaired social communication and at least two symptoms of repetitive behaviors or restricted interests[1]. ASD has a median prevalence of 1 out of 161 individuals in a study of worldwide data[2], with a higher diagnosis rate in some developed countries such as the United States[3].

Characterizing the neurobiology of ASD may eventually lead to improved diagnosis and clinical subgrouping, and the development of individually targeted treatment programs[4]. Although much of the neurobiology of ASD remains unknown, subtle alterations of brain structure appear to be involved (reviewed in ref. [5,6]). These include differences in total brain volume (children with ASD have shown a larger average volume[7–10]), as well as alterations of the medial and inferior frontal, anterior cingulate, superior temporal, and orbitofrontal cortices, and the caudate nucleus[5,6,11]. However, the results of structural magnetic resonance imaging (MRI) studies of ASD have often been inconsistent, potentially owing to (1) small study sample sizes in relation to subtle effects, (2) differences across studies in terms of clinical characteristics, age, comorbidity, and medication use, (3) methodological differences between studies, such as differences in hardware, software, and distinct data processing pipelines[12], and (4) the etiological and neurobiological heterogeneity of ASD, which exists as a group of different syndromes rather than a single entity[13].

In the ENIGMA (Enhancing Neuro-Imaging Genetics through Meta-Analysis) consortium (http://enigma.ini.usc.edu), researchers from around the world collaborate to analyze many separate data sets jointly, and to reduce some of the technical heterogeneity by using harmonized protocols for MRI data processing. A recent study by the ENIGMA consortium's ASD working group showed small average differences in bilateral cortical and subcortical brain measures between 1571 cases and 1650 healthy controls, in the largest study of brain structure in ASD yet performed[14]. Relative to controls, ASD patients had significantly lower volumes of several subcortical structures, as well as greater thickness in various cortical regions—mostly in the frontal lobes—and lower thickness of temporal regions. No associations of diagnosis with regional cortical surface areas were found[14].

Left–right asymmetry is an important aspect of human brain organization, which may be altered in various psychiatric and neurocognitive conditions, including schizophrenia, dyslexia, and ASD[15–17]. On a functional level, people with ASD demonstrate reduced leftward language lateralization more frequently than controls[18–20]. Resting-state functional MRI data of people with ASD have also shown a generally rightward shift of asymmetry involving various functional networks of brain regions[21]. In addition, people with ASD have a higher rate of left-handedness than the general population[20,22,23]. Furthermore, an electroencephalography study reported that infants at high risk for ASD showed more rightward than leftward frontal alpha asymmetry at rest[24].

Brain structural imaging studies have also reported altered hemispheric asymmetry in ASD. Diffusion imaging studies indicated reduced asymmetry of a variety of different white matter tract metrics[25–27], although in one study males with ASD lacked an age-dependent decrease in rightward asymmetry of network global efficiency, compared with controls[28]. A structural MRI study investigating gray matter reported lower leftward volume asymmetry of language-related cortical regions in ASD (i.e., *planum temporale*, Heschl's gyrus, posterior supramarginal gyrus, and parietal operculum), as well as greater rightward asymmetry of the inferior parietal lobule[29]. The volume and surface area of the fusiform gyrus also showed lower rightward asymmetry in ASD[30]. However, other studies did not find alterations of gray matter asymmetries in ASD[27,31].

Prior studies of structural brain asymmetry in ASD had sample sizes less than 128 cases and 127 controls. The previous ENIGMA consortium study of ASD[14] did not perform analyses of brain asymmetry, but reported bilateral effects only as strong as Cohen's $d = -0.21$ (for entorhinal thickness bilaterally)[14]. Comparable bilateral effect sizes were also found in ENIGMA consortium studies of other disorders[14,32–38]. If effects on brain asymmetry are similarly subtle, then prior studies of this aspect of brain structure in ASD were likely underpowered. Low power not only reduces the chance of detecting true effects, but also the likelihood that a statistically significant result reflects a true effect[39,40]. Therefore large-scale analysis is needed to determine whether, and how, structural brain asymmetry might be altered in ASD, to better describe the neurobiology of the condition.

Here, we made use of MRI data from 54 data sets that were collected across the world by members of the ENIGMA consortium's ASD Working Group, to perform the first highly powered study of structural brain asymmetry in ASD. Using a single, harmonized protocol for image analysis, we derived asymmetry indexes, $\text{AI} = (\text{Left} - \text{Right})/(\text{Left} + \text{Right})$, for multiple brain regional and global hemispheric measures, in up to 1778 individuals with ASD and 1829 typically developing controls. The AI is a widely used index in brain asymmetry studies[41,42].

Age and sex are known to affect cortical-[43] as well as subcortical asymmetries[44] in healthy individuals. In addition, a recent structural imaging study of roughly 500 individuals with ASD, and 800 controls, found that case–control differences of bilateral cortical thickness were greater in younger versus older individuals, whereas also being related to ASD symptom severity, and with larger differences in individuals with lower versus higher full-scale intelligent quotient (IQ) scores[45]. Other previous case–control MRI findings with respect to these indicators of clinical heterogeneity in ASD are also reviewed in that paper[45]. In the present study, we therefore carried out secondary analyses in which we tested brain asymmetries in relation to age- or sex-specific effects, IQ, and disorder severity. We also included an exploratory analysis of medication use.

## Results

**Significant associations of ASD with brain asymmetry.** Summary information for the data sets is in Table 1. Out of a total of 78 structural AIs that were investigated (Supplementary Tables 1–3), 10 showed a significant effect of diagnosis, which survived multiple testing correction (Table 2). Among these were seven regional cortical thickness AIs, including frontal regions (superior frontal, rostral middle frontal, medial orbitofrontal), temporal regions (fusiform, inferior temporal), and cingulate regions (rostral anterior, isthmus cingulate). Two cortical regional surface area AIs, namely of the medial- and lateral orbitofrontal cortex, were significantly associated with diagnosis (medial: $\beta = 0.006$, $t = 3.2$, $P = 0.0015$; lateral: $\beta = -0.005$, $t = -3.3$, $P = 0.0010$) (Table 2, Supplementary Table 2), as well as one subcortical volume AI, namely that of the putamen ($\beta = 0.00395$, $t = 3.4$, $P = 0.00069$) (Table 2, Supplementary Table 3).

Nominally significant effects of diagnosis on AIs (i.e., with $P < 0.05$, but which did not survive multiple comparison correction),

**Table 1 Characteristics of the different data sets of the ENIGMA ASD working group**

| Sample name | N total | N cases (M/F) | N controls (M/F) | Median age in years (range) | Scanner type | Field strength |
|---|---|---|---|---|---|---|
| ABIDE_CALTECH | 31 | 13/1 | 13/4 | 23.4 (17.5, 56.2) | Siemens Trio | 3 T |
| ABIDE_KKI | 21 | 7/0 | 11/3 | 10.6 (8.4, 12.8) | Philips Achieva | 3 T |
| ABIDE_LEUVEN_1 | 29 | 14/0 | 15/0 | 22 (18, 32) | Philips Interna | 3 T |
| ABIDE_LEUVEN_2 | 35 | 12/3 | 15/5 | 14.2 (12.1, 16.9) | Philips Interna | 3 T |
| ABIDE_MAX_MUN | 57 | 21/3 | 29/4 | 26 (7, 58) | Siemens Verio | 3 T |
| ABIDE_NYU | 186 | 68/11 | 81/26 | 13.6 (6.5, 39.1) | Siemens Allegra | 3 T |
| ABIDE_OHSU | 18 | 7/0 | 11/0 | 10 (8.2, 12.7) | Siemens Trio | 3 T |
| ABIDE_OLIN | 36 | 17/3 | 14/2 | 17 (10, 24) | Siemens Allegra | 3 T |
| ABIDE_PITT | 58 | 26/5 | 23/4 | 17 (9.3, 35.2) | Siemens Allegra | 3 T |
| ABIDE_SBL | 30 | 15/0 | 15/0 | 33.5 (20, 64) | Philips Interna | 3 T |
| ABIDE_SDSU | 37 | 14/1 | 16/6 | 14.8 (8.7, 37.7) | GE MR750 | 3 T |
| ABIDE_STANFORD | 40 | 16/4 | 16/4 | 9.4 (7.5, 12.9) | GR Signa | 3 T |
| ABIDE_TCD | 55 | 24/1 | 30/0 | 15.9 (9.3, 25.9) | Philips Achieva | 3 T |
| ABIDE_UM_1 | 130 | 50/14 | 43/23 | 12.3 (8.1, 20.9) | GE Signa | 3 T |
| ABIDE_UM_2 | 31 | 14/1 | 15/1 | 14.8 (11.1, 26.8) | GE Signa | 3 T |
| ABIDE_USM | 101 | 59/0 | 42/0 | 19.6 (8.2, 50.2) | Siemens Trio | 3 T |
| ABIDE_YALE | 55 | 20/8 | 19/8 | 12.8 (7, 17.8) | Siemens Magnetom | 3 T |
| ABIDEII-BNI | 58 | 29/0 | 29/0 | 43 (18, 64) | Philips Ingenia | 3 T |
| ABIDEII-EMC | 54 | 22/5 | 22/5 | 8.2 (6.2, 10.7) | GE MR750 | 3 T |
| ABIDEII-ETH | 37 | 13/0 | 24/0 | 22.3 (13.8, 30.7) | Philips Achieva | 3 T |
| ABIDEII-GU | 106 | 43/8 | 28/27 | 10.6 (8.1, 13.9) | Siemens TriTim | 3 T |
| ABIDEII-IP | 56 | 14/8 | 12/22 | 18.4 (6.1, 46.6) | Siemens TriTim | 1.5 T |
| ABIDEII-IU | 40 | 16/4 | 15/5 | 22 (17, 54) | Philips Achieva | 3 T |
| ABIDEII-KKI | 211 | 41/15 | 99/56 | 10.3 (8, 13) | Philips Achieva | 3 T |
| ABIDEII-KUL* | 28 | 28/0 | – | 24 (18, 35) | Philips Achieva | 3 T |
| ABIDEII-NYU_1 | 78 | 43/5 | 28/2 | 8.4 (5.2, 34.8) | Siemens Allegra | 3 T |
| ABIDEII-NYU_2* | 27 | 24/3 | – | 7 (5.1, 8.8) | Siemens Allegra | 3 T |
| ABIDEII-OHSU | 93 | 30/7 | 27/29 | 11 (7, 15) | Siemens Skyra | 3 T |
| ABIDEII-OILH | 59 | 20/4 | 20/15 | 23 (18, 31) | Siemens TriTim | 3 T |
| ABIDEII-SDSU | 57 | 26/7 | 22/2 | 13 (7.4, 18) | GE MR750 | 3 T |
| ABIDEII-TCD | 43 | 21/0 | 22/0 | 14.8 (10, 20) | Philips Achieva | 3 T |
| ABIDEII-UCD | 32 | 14/4 | 10/4 | 14.6 (12, 17.8) | Siemens TriTim | 3 T |
| ABIDEII-UCLA | 32 | 15/1 | 11/5 | 9.6 (7.8, 15) | Siemens TriTim | 3 T |
| ABIDEII-USM | 33 | 15/2 | 13/3 | 19.7 (9.1, 38.9) | Siemens TriTim | 3 T |
| Barcelona | 76 | 39/4 | 32/1 | 12.3 (7.2, 17.1) | Siemens Trio | 3 T |
| BRC | 52 | 19/0 | 33/0 | 15 (10, 18) | GE Signa HDx | 3 T |
| CMU | 27 | 11/3 | 10/3 | 27 (19, 40) | Siemens Magnetom | 3 T |
| Dresden | 45 | 18/3 | 20/4 | 31.2 (21.1, 56.8) | Siemens Trio | 3 T |
| FAIR | 85 | 36/7 | 27/15 | 11.6 (7.2, 15.9) | Siemens Magnetom | 3 T |
| FRANKFURT | 27 | 10/2 | 13/2 | 18 (18, 18) | Siemens Sonata | 1.5 T |
| FSM | 80 | 20/20 | 20/20 | 4.1 (1.8, 6) | GE Signa | 1.5 T |
| MRC | 148 | 74/0 | 74/0 | 25 (18, 45) | GE Signa HDx | 3 T |
| MYAD | 73 | 59/0 | 14/0 | 4.5 (1.5, 9) | Siemens symphony | 1.5 T |
| NIJMEGEN1_1 | 33 | 14/3 | 14/2 | 15.1 (12.3, 18) | Siemens Trio | 3 T |
| NIJMEGEN1_2* | 9 | – | 8/1 | 17.5 (13.4, 18.5) | Siemens Trio | 3 T |
| NIJMEGEN2 | 72 | 29/19 | 16/8 | 26 (18, 40) | Siemens Avanto | 1.5 T |
| NIJMEGEN3 | 95 | 37/4 | 45/9 | 9.7 (6.1, 12.3) | Siemens Avanto | 1.5 T |
| ParelladaHGGM | 66 | 33/2 | 30/1 | 13 (7, 18) | Philips Intera | 1.5 T |
| PITT_1 | 56 | 11/3 | 34/8 | 15 (8, 36) | Siemens Allegra | 3 T |
| PITT_2 | 90 | 39/6 | 39/6 | 15 (8, 36) | Siemens Allegra | 3 T |
| SAOPAULO | 35 | 15/0 | 20/0 | 11 (6, 19) | Philps | 3 T |
| TCD_1 | 50 | 28/0 | 22/0 | 14.9 (10, 21.8) | Philips Achieva | 3 T |
| TCD_2 | 39 | 17/0 | 22/0 | 15.9 (12, 25.9) | Philips Achieva | 3 T |
| TORONTO_1 | 219 | 82/23 | 57/57 | 11.9 (3.3, 20.8) | Siemens Trio | 3 T |
| TORONTO_2 | 203 | 107/43 | 29/24 | 11 (2.5, 21.7) | Siemens Trio | 3 T |
| UMCU_1 | 88 | 41/7 | 37/3 | 12.1 (7.1, 24.7) | Philips | 1.5 T |
| UMCU_2 | 9 | 6/0 | 2/1 | 11.8 (10, 12.6) | Philips | 1.5 T |
| **Total** | **3671** | **1833** | **1838** | | | |

*Excluded, as diagnosis would be completely confounded with random variable 'data set' in the analysis model. For the calculation of the totals (in bold) these data sets were also excluded

were observed for the fusiform surface area AI ($\beta = -0.005$, $t = -2.56$, $P = 0.010$) (Supplementary Table 2), pars orbitalis thickness AI ($\beta = -0.003$, $t = -2.26$, $P = 0.024$), posterior cingulate thickness AI ($\beta = -0.003$, $t = -2.1$, $P = 0.034$), superior temporal thickness ($\beta = -0.002$, $t = -1.97$, $P = 0.049$), and caudate nucleus volume ($\beta = 0.003$, $t = 2.24$, $P = 0.025$).

**Sensitivity analyses**. When we repeated the analysis after winsorizing outliers, the pattern of results remained the same (Supplementary Tables 4–6), except that a small change in P value for the effect of diagnosis on medial orbitofrontal surface area AI meant that it no longer survived false discovery rate (FDR) correction (Supplementary Table 5).

**Table 2 Linear mixed model results for regional AIs that survived multiple comparisons correction in the primary analysis**

| AI region | N cases/controls | t value diag | t value age | t value sex | P value diag | P value age | P value sex | Cohen's d (95% CI) |
|---|---|---|---|---|---|---|---|---|
| Fusiform thickness | 1704/1767 | 3.20 | −0.17 | 2.06 | 0.001 | 0.862 | 0.040 | 0.109 (0.04,0.18) |
| Inferior temporal thickness | 1703/1768 | 2.97 | 0.07 | 0.73 | 0.003 | 0.948 | 0.467 | 0.102 (0.03,0.17) |
| Isthmus cingulate thickness | 1699/1769 | −2.58 | 2.81 | 1.02 | 0.010 | 0.005 | 0.310 | −0.088 (−0.16,−0.02) |
| Medial orbitofrontal thickness | 1705/1769 | −3.47 | 4.23 | −0.31 | 0.001 | $2.36 \cdot 10^{-5}$ | 0.758 | −0.119 (−0.19,−0.05) |
| Rostral anterior cingulate thickness | 1700/1765 | −3.37 | −0.69 | 1.45 | 0.001 | 0.491 | 0.147 | −0.116 (−0.18,−0.05) |
| Rostral middle frontal thickness | 1707/1769 | −2.94 | −2.87 | 1.47 | 0.003 | 0.004 | 0.141 | −0.101 (−0.17,−0.03) |
| Superior frontal thickness | 1706/1771 | −3.92 | 0.48 | −1.38 | $8.86 \cdot 10^{-5}$ | 0.630 | 0.169 | −0.134 (−0.2,−0.07) |
| Lateral orbitofrontal surface area | 1701/1762 | −3.28 | 0.81 | −0.07 | 0.001 | 0.421 | 0.946 | −0.112 (−0.18,−0.05) |
| Medial orbitofrontal surface area | 1704/1763 | 3.17 | −0.27 | −0.87 | 0.002 | 0.787 | 0.385 | 0.109 (0.04,0.18) |
| Putamen | 1712/1763 | 3.40 | 5.03 | 1.77 | 0.001 | $5.21 \cdot 10^{-7}$ | 0.076 | 0.116 (0.05,0.18) |

aUnadjusted P values are shown. Note that this table only includes AIs for which the effect of diagnosis was significant after FDR correction. Results for all AIs are in supplementary information (Supplementary Tables 1–3)

When we added a non-linear effect for age, all of the 10 AIs that had shown significant effects of diagnosis in the primary analysis remained significant (Supplementary Tables 4–6).

When we excluded all individuals below 6 years of age, that may have been more difficult for FreeSurfer to segment, all AIs that had shown significant effects of diagnosis in the primary analysis remained significant, except for the isthmus cingulate thickness AI (Supplementary Tables 4–6). In addition, one new association with diagnosis, of the fusiform surface area AI, now surpassed the multiple testing correction threshold. These subtle changes of P values do not necessarily indicate that exclusion of younger ages improved signal to noise in the data.

When excluding all individuals aged 40 years or older, the pattern of significant results stayed the same (Supplementary Tables 4–6).

Finally, when analyzing only the subset of 3T-acquired data, two of the diagnosis effects from the primary analysis (i.e., inferior temporal- and isthmus cingulate thickness AI) were no longer significant after false discovery rate correction, but three other effects now became significant (i.e., superior temporal thickness AI, fusiform surface area AI, and caudate nucleus AI) (Supplementary Tables 4–6). Again, slight changes in significance levels are expected when changing the sample, and do not necessarily indicate systematic differences of 3 T and 1.5 T data with respect to case–control asymmetry differences.

**Magnitudes and directions of asymmetry changes.** Cohen's *d* effect sizes of the associations between AIs and diagnosis, as derived from the primary analysis, are visualized in Fig. 1. Effect sizes were low, ranging from −0.13 (superior frontal thickness AI) to 0.12 (Putamen AI) (Table 2, Supplementary Tables 1–3).

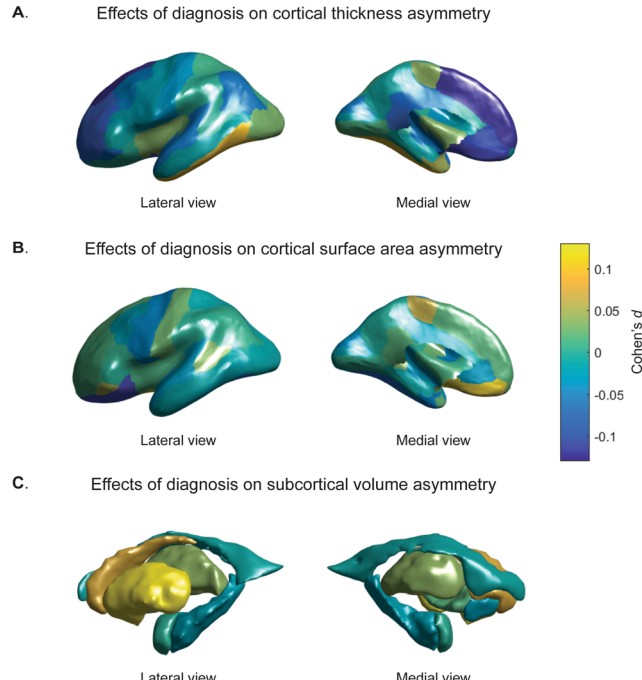

**A.** Effects of diagnosis on cortical thickness asymmetry

Lateral view    Medial view

**B.** Effects of diagnosis on cortical surface area asymmetry

Lateral view    Medial view

**C.** Effects of diagnosis on subcortical volume asymmetry

Lateral view    Medial view

Cohen's d
0.1
0.05
0
−0.05
−0.1

**Fig. 1** Cohen's *d* effect sizes of the associations between diagnosis and AIs. **a** regional cortical thickness measures, **b** cortical surface areas, **c** subcortical volumes. Values are overlaid on left hemisphere inflated brains. Positive Cohen's *d* values (yellow) indicate mean shifts towards greater leftward or reduced rightward asymmetry in cases, and negative Cohen's *d* values (blue) indicate mean shifts towards greater rightward asymmetry or reduced leftward asymmetry in individuals with ASD

All of the cortical AIs with significant effects of diagnosis in the primary analysis showed decreased asymmetry in ASD compared with controls, i.e., the AIs were closer to zero in individuals with ASD than in controls, regardless of whether the region was on average leftward or rightward asymmetrical in controls (Table 3). However, the putamen showed increased asymmetry in ASD (mean AI controls = 0.011, mean AI cases = 0.012) (Table 3).

Five of the seven significant changes in regional cortical thickness asymmetry involved left-sided decreases accompanied by right-sided increases of thickness (Table 3). For the other two significant effects on regional thickness asymmetry (the fusiform and inferior temporal cortex), thickness was decreased bilaterally in ASD, but more so in the right than the left hemisphere. For the significant changes in surface area asymmetry (lateral orbitofrontal and medial orbitofrontal cortex), surface area was altered in opposite directions in ASD in the two hemispheres, thus resulting in altered asymmetry (Table 3). Finally, the putamen showed a bilateral decrease in volume in ASD that was more pronounced on the right, resulting in altered asymmetry (Table 3).

**Age or sex interaction effects**. The distributions of age and sex across all data sets are plotted in Supplementary Fig. 1. In secondary analysis of interaction effects, there was only one significant sex:diagnosis interaction effect after FDR correction, for the rostral anterior cingulate thickness AI (Supplementary Tables 7–9). This AI had shown a significant effect of diagnosis in the primary analysis. In analysis within the sexes separately, this AI was associated with diagnosis in males ($P = 1.4 \times 10^{-5}$) but not females ($p = 0.165$) (Supplementary Table 7). For all of the AIs, which showed significant effects of diagnosis in the primary analysis, adding sex:diagnosis interaction terms did not change the pattern of significant main effects of diagnosis, after FDR correction (Supplementary Tables 7–9).

There were no significant age:diagnosis interaction effects after FDR correction (Supplementary Tables 10–12). In general, for AIs which showed significant effects of diagnosis in the primary analysis, adding age:diagnosis interaction terms largely reduced the significance of the main effects of diagnosis, even though the age:diagnosis interaction terms were not significant (all $P > 0.05$) (Supplementary Tables 10–12). However, adding these interaction terms also increased the AIC and BIC scores compared with the primary analysis models without these terms, indicating poorer model fit when including these non-significant interaction terms (Supplementary Tables 10–12).

**Exploratory analysis of IQ**. The distributions of IQ within individuals with ASD and controls are shown in Supplementary Fig. 2. Out of the 10 AIs that showed significant case–control differences in the primary analysis, only one showed an association with IQ within individuals with ASD (uncorrected $P < 0.05$; Supplementary Table 13). This was the rostral anterior cingulate thickness AI ($\beta = 0.00019$, $t = 2.49$, $p = 0.013$). The positive direction of this effect indicates that primarily those ASD individuals with lower IQ show reduced leftward asymmetry of the rostral anterior cingulate thickness. This regional asymmetry had also shown a significant sex*diagnosis interaction (see above). For this specific regional AI, we therefore added age:IQ, sex:IQ and age:sex:IQ interactions to the model, but none of these terms were significant (all uncorrected $P > 0.05$).

Within controls, only the superior frontal thickness AI was associated with IQ at uncorrected P < 0.05 (Supplementary Table 13) ($\beta = -0.00012$, $t = -3.41$, $p = 0.001$). This effect suggests that controls with lower IQ show relatively increased

**Table 3 Directions of asymmetry changes in cases versus controls**

| AI region | mean AI ± SD in controls | mean AI ± SD in ASD | Cohen's d (95% CI) Left hemisphere | Cohen's d (95% CI) Right hemisphere | Controls | ASD |
|---|---|---|---|---|---|---|
| Fusiform thickness | −0.004 ± 0.02 | −0.002 ± 0.03 | −0.152 (−0.22, −0.09) | −0.22 (−0.29, −0.15) | Rightward | Decreased |
| Inferior temporal thickness | −0.006 ± 0.03 | −0.003 ± 0.03 | −0.13 (−0.2, −0.06) | −0.194 (−0.26, −0.13) | Rightward | Decreased |
| Isthmus cingulate thickness | 0.012 ± 0.04 | 0.007 ± 0.04 | −0.014 (−0.08, 0.05) | 0.07 (0, 0.14) | Leftward | Decreased |
| Medial orbitofrontal thickness | 0.003 ± 0.04 | −0.001 ± 0.04 | −0.009 (−0.08, 0.06) | 0.087 (0.02, 0.15) | Leftward | Reversed |
| Rostral anterior cingulate thickness | 0.016 ± 0.05 | 0.011 ± 0.05 | −0.116 (−0.18, −0.05) | 0.009 (−0.06, 0.08) | Leftward | Decreased |
| Rostral middle frontal thickness | 0.01 ± 0.03 | 0.008 ± 0.03 | −0.016 (−0.08, 0.05) | 0.046 (−0.02, 0.11) | Leftward | Decreased |
| Superior frontal thickness | 0.006 ± 0.02 | 0.005 ± 0.02 | −0.058 (−0.12, 0.01) | 0.018 (−0.05, 0.08) | Leftward | Decreased |
| Lateral orbitofrontal surface area | 0.014 ± 0.05 | 0.009 ± 0.05 | −0.027 (−0.09, 0.04) | 0.032 (−0.04, 0.1) | Leftward | Decreased |
| Medial orbitofrontal surface area | −0.012 ± 0.06 | −0.004 ± 0.06 | 0.076 (0.01, 0.14) | −0.001 (−0.07, 0.07) | Rightward | Decreased |
| Putamen | 0.011 ± 0.03 | 0.012 ± 0.04 | −0.029 (−0.1, 0.04) | −0.095 (−0.16, −0.03) | Leftward | Increased |

The raw means and standard deviations are indicated for each AI that showed a significant effect of diagnosis in the primary analysis. Also the Cohen's d effect sizes for left and right hemispheric measures are indicated (i.e., when left or right hemispheric measures were analyzed as dependent variables). In addition, the average direction of asymmetry in controls (derived from the raw mean AI) and its change in ASD is shown. Positive AI values indicate leftward asymmetry, negative AI values indicate rightward asymmetry

asymmetry of superior frontal thickness, although this was post hoc, exploratory analysis without multiple testing correction.

**Analysis of autism diagnostic observation schedule (ADOS) severity scores**. The distributions of ADOS severity scores are plotted in Supplementary Fig. 2. Out of the AIs that showed significant case–control differences in the primary analysis, only the isthmus cingulate thickness AI showed an association (uncorrected $P < 0.05$) with the ADOS severity score ($\beta = 0.0041$, $t = 2.6$, $p = 0.011$) (Supplementary Table 14). The positive direction of the effect suggests that primarily cases with low ASD severity have reduced leftward asymmetry of this regional thickness.

**Medication use**. We found no significant effects of medication use (all uncorrected $P > 0.05$) (Supplementary Table 15).

## Discussion

In this, the largest study to date of brain asymmetry in ASD, we mapped differences in brain asymmetry between participants with ASD and controls, in a collection of 54 international data sets via the ENIGMA Consortium. We had 80% statistical power to detect Cohen's $d$ effect sizes in the range of 0.12–0.13. We found significantly altered asymmetries of seven regional cortical thickness asymmetries in ASD compared with controls, predominantly involving medial frontal, orbitofrontal, inferior temporal, and cingulate regions. The magnitude of all regional thickness asymmetries was decreased in ASD compared with controls, whether it was reduced leftward, reduced rightward, or reversed average asymmetry. Rightward asymmetry of the medial orbitofrontal surface area was also decreased in individuals with ASD, as was leftward asymmetry of the lateral orbitofrontal surface area. In addition, individuals with ASD showed an increase in leftward asymmetry of putamen volume, compared with controls.

Previous MRI studies of cerebral cortical asymmetries in ASD, based on much smaller data sets, and using diverse methods for image analysis, suggested variable case–control differences[29,30], or no differences[27,31]. Our findings partly support a previously reported, generalized reduction of leftward asymmetry[29], as six of the nine significantly altered cortical regional asymmetries (thickness or surface area) involved decreased leftward asymmetries. However, three of the nine significantly altered cortical regional asymmetries involved shifts leftwards in ASD, either driven by a more prominent increase on the left side in ASD (i.e., medial orbitofrontal surface area), or by more prominent right-than left-side decreases in ASD (i.e., fusiform- and inferior temporal thickness). Thus, the directional change of asymmetry can depend on the specific region, albeit that the overall magnitude of asymmetry is most likely to be reduced in ASD.

The significant associations of diagnosis with asymmetry in the present study were all weak (Cohen's $d = -0.13$–0.12), indicating that altered structural brain asymmetry is unlikely to be a useful predictor for ASD. Prior studies using smaller samples were underpowered in this context. However, the effect sizes were comparable to those reported by recent, large-scale studies of bilateral disorder-related changes in brain structure, in which asymmetry was not studied, including for ASD[14] as well as attention-deficit hyperactivity disorder (ADHD)[38,46], schizophrenia[37], obsessive compulsive disorder (OCD)[32,33], posttraumatic stress disorder[34], and major depressive disorder[35,36]. It has become increasingly clear that anatomical differences between ASD and control groups are very small relative to the large within-group variability that is observed[47].

Our findings may inform understanding of the neurobiology of ASD. Multi-regional reduction of cortical thickness asymmetry in ASD fits with the concept that laterality is an important organizing feature of the healthy human brain for multiple aspects of complex cognition, and is susceptible to disruption in disorders (e.g., [16,48]). Left–right asymmetry facilitates the development of localized and specialized modules in the brain, which can then have dominant control of behavior[49,50]. Notably, some of the cortical regions highlighted here are involved in diverse social cognitive processes, including perceptual processing (fusiform gyri), cognitive and emotional control (anterior cingulate), and reward evaluation (orbitofrontal cortex, ventral striatum)[51]. However, the roles of these brain structures are by no means restricted to social behavior. As we found altered asymmetry of various additional regions, our findings suggest broader disruption of lateralized neurodevelopment as part of the ASD phenotype. We note that many of the regions that showed significant case–control differences in asymmetry, including medial frontal, anterior cingulate, and inferior temporal regions, overlap with the default mode network (DMN). The DMN comprises various cortical regions located in temporal (medial and lateral), parietal (medial and lateral) and prefrontal (medial) cortices[52]. DMN network organization has shown evidence for differences in ASD[11,53–55], including alterations in functional laterality[56]. Our findings may therefore further support a role of altered lateralization of the DMN in ASD, warranting further investigations in this direction.

The medial orbitofrontal cortex was the only region that showed significantly altered asymmetry of both thickness and surface area in ASD, suggesting that disrupted laterality of this region might be particularly important in ASD. The orbitofrontal cortex may be involved in repetitive and stereotyped behaviors in ASD, owing to its roles in executive functions[57]. Prior studies have reported lower cortical thickness in the left medial orbitofrontal gyrus in ASD[58], altered patterning of gyri and sulci in the right orbitofrontal cortex[59], and altered asymmetry in frontal regions globally[25,31]. These studies were in much smaller sample sizes than used here.

As regards the fusiform cortex, a previous study by Dougherty et al.[30] reported an association between higher ASD symptom severity and increased rightward surface area asymmetry, but not thickness asymmetry. The fusiform gyrus is involved in facial perception and memory among other functions, which are important for social interactions[60]. Here we report an asymmetry change in fusiform thickness in ASD that was significant after multiple testing correction, but there was also a nominally significant rightward change of surface area asymmetry in ASD (i.e., that did not survive multiple testing correction). This underlines that separate analyses of regional cortical thickness and surface area are well motivated, as they can vary relatively independently[61].

The altered volume asymmetry of the putamen in ASD may be related to its role in repetitive and restricted behaviors in ASD. One study reported that differences in striatal growth trajectories were correlated with circumscribed interests and insistence on sameness[62]. The striatum is connected with lateral and orbitofrontal regions of the cortex via lateral–frontal–striatal reward and top-down cognitive control circuitry that might be dysfunctional in ASD[63]. For example, individuals with ASD have shown decreased activation of the ventral striatum and lateral inferior/orbitofrontal cortex during outcome anticipation, and of dorsal striatum and lateral–frontal regions during sustained attention and inhibitory control, compared with typically developing controls[11,55,63].

Although the reasons for asymmetrical alterations in many of the structures implicated here are unclear, our findings suggest

altered neurodevelopment affecting these structures in ASD. Further research is necessary to clarify the functional relevance and relationships between altered asymmetry and ASD. The findings we report in this large–scale study sometimes did not concur with prior, smaller studies. This may be owing to limited statistical power in the earlier studies, whereas low power reduces the likelihood that a statistically significant result reflects a true effect[40]. However, the cortical atlas that we used did not have perfect equivalents for regions defined in many of the earlier studies, and we did not consider gyral/sulcal patterns, or gray matter volumes as such. Furthermore, discrepancies with earlier studies may be related to age differences, and differences in clinical features of the disorder arising from case recruitment and diagnosis.

We included subjects from the entire ASD severity spectrum, with a broad range of ages, IQs, and of both sexes. Only one effect of diagnosis on regional asymmetry was influenced by sex, i.e., the rostral anterior cingulate thickness asymmetry, which was altered in males but not in females. This same regional asymmetry was primarily altered in lower versus higher IQ cases. This may therefore be an alteration of cortical asymmetry that is relatively specific to an ASD subgroup, i.e., lower-performing males. In controls, a different asymmetry (i.e., superior frontal thickness AI) showed a nominally significant association with IQ, which may point to different brain-IQ associations in ASD and controls. However, we cannot make strong interpretations based on these exploratory, secondary analyses without multiple testing correction.

As regards symptom severity, thickness asymmetry of the isthmus of the cingulate was associated with the ADOS score, such that the lower severity cases tended to have the most altered asymmetry. Again, this post hoc finding remains tentative in the context of multiple testing, and is reported here for descriptive purposes only. It is clear that most of the AIs that showed significant changes in ASD were not correlated with ADOS scores.

We found no evidence that medication use affected any of the asymmetries altered in ASD, although our medication variable was rudimentary. The role of specific medication usage should be investigated in future studies. As mentioned above, data on comorbidities were only available for 54 of the ASD subjects, precluding a high-powered analysis of this issue. This is a limitation of the study. We did not analyze handedness in the present study, as this had no significant effect on the same brain asymmetry measures as analyzed here, in studies of healthy individuals comprising more than 15,000 participants[43,44].

In contrast to some prior studies of ASD, we did not adjust for IQ as a covariate effect in our main, case–control analysis. Rather, we carried out post hoc analysis of possible associations between IQ and brain asymmetries, separately in cases and controls. This was because lower average IQ was clearly part of the ASD phenotype in our total combined data set (Supplementary Fig. 1D), so that including IQ as a confounding factor in case–control analysis might have reduced the power to detect an association of diagnosis with asymmetry. This would occur if underlying susceptibility factors contribute both to altered asymmetry and reduced IQ, as part of the ASD phenotype.

The Desikan–Killiany atlas[64] was derived from manual segmentations of sets of reference brain images. Accordingly, the mean regional asymmetries in our samples partly reflect left–right differences present in the reference data set used to construct the atlas. For detecting cerebral asymmetries with automated methods, some groups have chosen to work from artificially created, left–right symmetrical atlases, e.g., ref. [65]. However, our study was focused on comparing relative asymmetry between groups. The use of a 'real-world' asymmetrical atlas had the advantage that regional identification was likely to be more accurate for

structures that are asymmetrical both in the atlas and, on average, in our data sets. By defining the regions of interest in each hemisphere based on each hemisphere's own particular features, such as its sulcal and gyral geometry, we could then obtain the corresponding relationships between hemispheres. To this end, we used data from the automated labeling program within FreeSurfer for subdividing the human cerebral cortex. The labeling system incorporates hemisphere-specific information on sulcal and gyral geometry with spatial information regarding the locations of brain structures, and shows a high accuracy when compared with manual labeling results[64]. Thus, reliable measures of each region can be extracted for each subject, and regional asymmetries then accurately assessed.

Although a single image analysis pipeline was applied to all data sets, heterogeneity of imaging protocols was a feature of this study. There were substantial differences between data sets in the average asymmetry measured for some regions, which may be owing in part to different scanner characteristics, as well as differences in patient profiles. We corrected for 'data set' as a random effect in the analysis, and sensitivity analysis based on the subset of 3 T acquired data showed similar results to the primary analysis. However, it is possible that between-data set variability resulted in reduced statistical power, relative to a hypothetical, equally sized, single-center study. In reality, no single centre has been able to collect such large samples alone. As long as researchers publish many separate papers based on single data sets, collected in particular ways, the field overall has the same problem. In this case, multi-centre studies can better represent the real-world heterogeneity, with more generalizable findings than single-centre studies[66]. The primary purpose of our study, based on 54 data sets that were originally collected as separate studies, was to assess the total combined evidence for effects over all of these data sets, whereas allowing for heterogeneity between data sets through the use of random intercepts, and finally adding sensitivity and secondary analyses with respect to relevant variables.

The cross-sectional design limits our capacity to make causal inferences between diagnosis and asymmetry. ASD is highly heritable, with meta-analytic heritability estimates ranging from 64 to 91%[67]. Likewise, some of the brain asymmetry measures examined here have heritabilities as high as roughly 25%[43,44]. Future studies are required to investigate shared genetic contributions to ASD and variation in brain structural asymmetry. These could help to disentangle cause-effect relations between ASD and brain structural asymmetry. Given the high comorbidity of ASD with other disorders, such as ADHD, OCD, and schizophrenia[68], cross-disorder analyses incorporating between-disorder genetic correlations may be informative.

In conclusion, large-scale analysis of brain asymmetry in ASD revealed primarily cortical thickness effects, but also effects on orbitofrontal cortex asymmetry, and putamen asymmetry, which were significant but very small. Our study illustrates how high-powered and systematic studies can yield much needed clarity in human clinical neuroscience, where prior smaller and more methodologically diverse studies were inconclusive.

## Methods

**Data sets.** Structural MRI data were available for 57 different data sets (Table 1). Three data sets comprising either cases only, or controls only, were removed in this study (Table 1), as our analysis model included random intercepts for 'data set' (below), and diagnosis was fully confounded with data set for these three. The remaining 54 data sets comprised 1778 people with ASD ($N = 1504$ males; median age = 13 years; range = 2–64 years) and 1829 typically developing controls ($N = 1400$ males; median age = 13 years; range = 2–64 years).

All data sets were collected during the period when DSM-IV and DSM-IV-TR were the common classification systems, between 1994 and 2013, and the clinical diagnosis of ASD was made according to DSM-IV criteria[69]. The data sets were

collected in a variety of different countries, and intended originally as separate studies. Nonetheless, all subjects were diagnosed based on clinical diagnosis by a clinically experienced and board certified physician/psychiatrist/psychologist. This was a criterion for admission of a data set into the ENIGMA-ASD database. For each of the 54 data sets, all relevant ethical regulations were complied with, and appropriate informed consent was obtained for all individuals.

Total scores from the Autism Diagnostic Observation Schedule-Generic (ADOS), a standardized instrument commonly used in autism diagnosis[70], were available for a majority of cases ($N = 878$). Cases from the entire ASD spectrum were included, but only 66 cases had IQ below 70 (cases: mean IQ = 104, SD = 19, min = 34, max = 149; see Supplementary Fig. 1D). The presence/absence of comorbid conditions had been recorded for 519 of the cases, but only 54 cases showed at least one comorbid condition (which could be ADHD, OCD, depression, anxiety, and/or Tourette's syndrome[14]). Numbers related to DSM-IV subtypes of ASD were not collated by the ENIGMA ASD working group, as this subtyping scheme has been dropped from DSM-V due to low reliability[71].

There was not a homogeneous assessment/recruitment process for controls across the 54 data sets, but the overwhelming majority were typically developing/healthy at the time of MRI, and no controls showed features that might have met criteria for a diagnosis of ASD. Only 19 controls had IQ > 70. In these subjects the exclusion of ASD diagnosis was performed by a senior child psychiatrist/physician. Eighteen of these were from the FSM data set and were clinically diagnosed with idiopathic intellectual disability. Amongst all controls the mean IQ was 112, SD = 15, min = 31, max = 149; see Supplementary Fig. 1D.

**Structural MRI.** Structural T1-weighted brain MRI scans were acquired at each study site. As shown in Table 1, images were acquired using different field strengths (1.5 T or 3 T) and scanner types. Each site used harmonized protocols from the ENIGMA consortium (http://enigma.ini.usc.edu/protocols/imaging-protocols) for data processing and quality control. The data used in the current study were thickness and surface area measures for each of 34 bilaterally paired cortical regions, as defined with the Desikan–Killiany atlas[64], as well as the average cortical thickness and total surface area per entire hemisphere. In addition, left and right volumes of seven bilaterally paired subcortical structures, plus the lateral ventricles, were analyzed. Cortical parcellations and subcortical segmentations were performed with the freely available and validated software FreeSurfer (versions 5.1 or 5.3)[72], using the default 'recon-all' pipeline, which also incorporates renormalization. Parcellations of cortical grey matter regions and segmentations of subcortical structures were visually inspected following the standardized ENIGMA quality control protocol ((http://enigma.ini.usc.edu/protocols/imaging-protocols). Exclusions on the basis of this quality control resulted in the sample sizes mentioned above (see Data sets). In briefly, cortical segmentations were overlayed on the T1 image of each subject. Web pages were generated with snapshots from internal slices, as well as external views of the segmentation from different angles. All sites were provided with the manual on how to judge these images, including the most common segmentation errors. For subcortical structures, the protocol again consisted of visually checking the individual images, plotted from a set of internal slices. Volume estimates derived from poorly segmented structures (i.e., where tissue labels were assigned incorrectly) were excluded from each site's data sets and subsequent analyses. In addition, any data points exceeding 1.5 times the interquartile range, as defined per site and diagnostic group, were visually inspected (3D). When identified as error, all values from the affected regions were excluded from further analysis.

**Asymmetry measures.** Separately for each structural measure and individual subject, left ($L$) and right ($R$) data were used in R (version 3.5.3) to calculate an asymmetry index (AI) with the following formula: $AI = (L − R)/(L + R)$. Distributions of each of the AIs are plotted in Supplementary Fig. 2. Note that AIs do not necessarily scale with $L$, $R$, or brain size, owing to their denominators.

**Linear mixed effects random-intercept model mega-analysis.** Model: linear mixed effects models were fitted separately for each cortical regional surface and thickness AI, as well as the total hemispheric surface area and mean thickness AI, and the subcortical volume AIs. This was accomplished by means of mega-analysis incorporating data from all 54 data sets, using the 'nlme' package in R[73]. All models included the same fixed- and random effects, and had the following formulation:

$$AI = diagnosis + age + sex + random(= dataset)$$

where AI reflects the AI of a given brain structure, and diagnosis ('controls' (= reference), 'ASD'), sex ('males' (= reference), 'females') and data set were coded as factor variables, with data set having 54 different categories. Age was coded as a numeric variable.

The Maximum Likelihood method was used to fit the models. Subjects were omitted if data were missing for any of the predictor variables (method = na.omit). The *ggplot2* package in R was used to visualize residuals (Supplementary Figs. 3–5). Collinearity of predictor variables was assessed using the 'usdm' package in R (version 3.5.3.).

**Significance.** Significance was assessed based on the $P$ values for the effects of diagnosis on AIs. The FDR[74] was estimated separately for the 35 cortical surface area AIs (i.e., 34 regional AIs and one hemispheric total AI) and the 35 cortical thickness AIs, and again for the seven subcortical structures plus lateral ventricles, each time with a FDR threshold of 0.05. Correlations between AI measures were calculated using Pearson's $R$ and visualized using the 'corrplot' package in R (Supplementary Figs. 6–8). Most pairwise correlations between AIs were low, with only 33/78 pairwise correlations either lower than $-0.3$ or $>0.3$, with a minimum $R = -0.351$ between the inferior parietal surface area AI and supramarginal surface area AI, and maximum $R = 0.487$ between the cuneus surface area AI and pericalcarine surface area AI.

**Cohen's *d* effect sizes.** The $t$-statistic for the factor 'diagnosis' in each linear mixed effects model was used to calculate Cohen's $d$[75], with

$$d = \frac{t * (n1 + n2)}{\sqrt{(n1 * n2)} * \sqrt{df}} \quad (1)$$

where n1 and n2 are the number of cases and controls, and $df$ the degrees of freedom.

The latter was derived from the lme summary table in R, but can also be calculated using $df = obs - (x1 + x2)$, where obs equals the number of observations, $x1$ the number of groups and $x2$ the number of factors in the model.

The 95% confidence intervals for Cohen's $d$ were calculated using 95% CI = $d \pm$ 1.96 SE, with the standard error (SE) around Cohen's $d$ calculated according to:

$$SE = \sqrt{\frac{n1 + n2}{n1 * n2} + \frac{d^2}{2 * (n1 + n2 - 2)}} \quad (2)$$

For visualization of cerebral cortical results, Cohen's $d$ values were loaded into Matlab (version R2016a), and 3D images of left hemisphere inflated cortical and subcortical structures were obtained using FreeSurfer-derived ply files.

**Power analyses.** As each linear model included multiple predictor variables, the power to detect an effect of diagnosis on AI could not be computed exactly, but we obtained an indication of the effect size that would be needed to provide 80% power, had we been using simple $t$ tests and Bonferroni correction for multiple testing, using the 'pwr' command in R. For this purpose, a significance level of 0.0014 (i.e., 0.05/35) was set in the context of multiple testing over the regional and total cortical surface area AIs ($N = 35$) or thickness AIs ($N = 35$), and 0.00625 (i.e., 0.05/8) for seven subcortical volume plus lateral ventricle AIs ($N = 8$). This showed that a difference of roughly Cohen's $d = 0.13$ would be detectable with 80% power in the cortical analyses, and Cohen's $d = 0.12$ in the subcortical analyses.

**Sensitivity analyses.** No outliers were removed for the primary analysis, but to confirm that results were not dependent on outliers, all analyses were repeated after having winsorized using a threshold of $k = 3$, for each AI measure separately. That is to say, the two highest and two lowest values were assigned the value of the third highest or lowest value, respectively, separately per AI. This threshold was chosen after visual inspection of frequency histograms.

The relationships between AIs and age showed no overt non-linearity (Supplementary Figs. 9–11), so no polynomials for age were incorporated in the models for primary analysis. However, analyses were repeated using an additional non-linear term for age, to check whether this choice had affected the results. As Age and Age$^2$ are highly correlated, we made use of the poly()-function in R for these two predictors, which created a pair of uncorrelated variables to model age effects (so-called orthogonal polynomials)[76], where one variable was linear and one non-linear.

As our data included participants as young as 1.5 years of age, and segmentation of very young brains might be especially challenging for the FreeSurfer algorithms, we also repeated our primary analysis excluding all individuals aged below 6 years ($N = 64$ controls, $N = 113$ cases), to assess whether they might have impacted the findings substantially (although these had passed the same quality control procedures as all other data sets, and FreeSurfer segmentation in preschoolers is generally of good quality, even before visual QC[77]).

As adults aged over 40 years were relatively sparsely represented, we also repeated the primary analysis after removing any individuals aged ≥40, in case modeling of age as a continuous predictor might have been unduly affected by these individuals. (In addition, see below for further analysis of age, for the purposes of subset and interaction analyses).

Finally, we repeated the primary analysis using only the subset of 3 T acquired data (45 out of 54 data sets), to test for possible sensitivity to this technical variable. The sample was reduced from 1778 cases and 1829 controls in the primary analysis to 1467 cases and 1574 controls in the 3T-only analysis.

**Directions of asymmetry changes.** For any AIs showing significant effects of diagnosis in the primary analysis, linear mixed effects modeling was also performed on the corresponding L and R measures separately, to understand the unilateral changes involved. The models included the same terms as were used in the main analysis of AIs (i.e., diagnosis, age and sex as fixed factors, and data set as random

factor). Again, the Cohen's *d* effect sizes for diagnosis were calculated based on the *t*-statistics. The raw mean AI values were calculated separately in controls and cases, to describe the reference direction of healthy asymmetry in controls, and whether cases showed reduced, increased, or reversed asymmetry relative to controls.

**Age- or sex-specific effects.** For all AIs, we carried out secondary analyses including age ∗ diagnosis and sex ∗ diagnosis interaction terms, in separate models. The models were as follows: AI = diagnosis + age + sex + age ∗ diag + random (= data set), and AI = diagnosis + age + sex + sex ∗ diag + random (= data set).

In addition, we separated the data into two subsets by age, i.e., children < 18 years and adults ≥ 18 years (using the same criteria as van Rooij et al. 2018), or else by sex (males, females). Models were then fitted separately for each AI within each subset, i.e., within each age subset AI = diagnosis + sex + random (= data set), and within each sex subset AI = diagnosis + age + random (= data set).

**Analysis of IQ.** For each AI that showed a significant effect of diagnosis in the primary analysis, we carried out exploratory analyses of IQ in cases and controls separately, whereby IQ (as a continuous variable) was considered as a predictor variable for the AI, so that AI = IQ + age + sex + random (= data set). This was done to understand whether individual differences in asymmetry might relate to IQ, and whether such relations might be specific to ASD.

**ADOS severity score.** For each AI that showed a significant effect of diagnosis in the primary analysis, a within-case-only analysis was performed incorporating symptom severity based on ADOS score as a predictor variable for AI: AI = ADOS + age + sex + random (= data set). This was to understand whether the observed asymmetry changes in cases were dependent on ASD severity. ADOS scores were first adjusted using $\log_{10}$ transformation to reduce skewing.

**Analysis of medication use.** Data on medication use (i.e., current use of psychiatric treatment drugs prescribed for ASD or comorbid psychiatric conditions) was available for 832 individuals with ASD, of which 214 were categorized as medication users. For each AI that showed a significant effect of diagnosis in the primary analysis, a linear mixed model analysis was performed within-cases only, AI = medication + age + sex + random (= data set). 'Medication' was coded as a binary variable (0 = no medication, 2 = medication).

**Reporting summary.** Further information on research design is available in the Nature Research Reporting Summary linked to this article.

## Data availability

This study made use of 54 separate data sets collected around the world, under a variety of different consent procedures and regulatory bodies, during the past 25 years. Requests to access the data sets will be considered in relation to the relevant consents, rules and regulations, and can be made via the ENIGMA consortium's ASD working group http://enigma.ini.usc.edu/ongoing/enigma-asd-working-group/

## Code availability

The R scripts used to generate the results of the current study are included (Supplementary Software 1).

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

## Acknowledgements

We thank the participants of all studies who have contributed data to the ENIGMA-ASD working group (http://enigma.ini.usc.edu/ongoing/enigma-asd-working-group/)[14]. This research was funded by the Max Planck Society (Germany). This study was further supported by the ENIGMA Center for Worldwide Medicine, Imaging & Genomics grant (NIH U54 EB020403) to Paul Thompson, and further supported by the Innovative Medicines Initiative Joint Undertaking under grant agreement number 115300 (EU-AIMS) and 777394 (AIMS-2-TRIALS), resources of which are composed of financial contribution from the European Union's Seventh Framework Programme and Horizon2020 programs and the European Federation of Pharmaceutical Industries and Associations (EFPIA) companies' in-kind contribution. The Canadian samples were collected as part of the POND network funded by the Ontario Brain Institute (grant IDS-I l-02 to Anagnostou / Lerch). Thanks to Derrek Hibar for helping to generate Fig. 1.

## Author contributions

P.T., S.E.F., D.v.R., J.B., and C.F. conceived the study. E.A., C.A., G.A., M.B., G.B.F., S.C., R.C., E.D., C.D., A.D.M., I.D., F.L.S.D., S.D., C.E., S.E., D.F., J.F., X.F., J.F., D.L.F., C.M.F., L.G., I.G, S.H., L.H., N.J., M.J., J.J., J.A.K., L.L., J.P.L., B.L., M.M.M., J.M., F.M., C.M.M., D.G.M.M., K.O., B.O., M.P., O.P., A.R., P.R., K.R., D.S., M.J.T., M.T., G.L.W., and F.Z. recruited, assessed and scanned the subjects, and performed MRI processing and quality control. D.C.G., S.E.M., and X.Z.K. contributed expertize on imaging and/or statistical analysis. D.v.R. organized the database. M.C.P. performed further quality control and the statistical analysis. M.C.P. and C.F. wrote the manuscript. All authors contributed edits and approved the content of the manuscript. C.F. directed the study.

## Competing interests

Dr. Anagnostou has served as a consultant or advisory board member for Roche and Takeda; she has received funding from the Alva Foundation, Autism Speaks, Brain Canada, the Canadian Institutes of Health Research, the Department of Defense, the National Centers of Excellence, NIH, the Ontario Brain Institute, the Physicians' Services Incorporated (PSI) Foundation, Sanofi-Aventis, and SynapDx, as well as in-kind research support from AMO Pharma; she receives royalties from American Psychiatric Press and Springer and an editorial honorarium from Wiley. Her contribution is on behalf of the POND network. Dr. Arango has served as a consultant for or received honoraria or grants from Acadia, Abbott, Amgen, CIBERSAM, Fundación Alicia Koplowitz, Instituto de Salud Carlos III, Janssen-Cilag, Lundbeck, Merck, Instituto de Salud Carlos III (co-financed by the European Regional Development Fund "A way of making Europe," CIBERSAM, the Madrid Regional Government [S2010/BMD-2422 AGES], the European Union Structural Funds, and the European Union Seventh Framework Programmeunder grant agreements FP7-HEALTH-2009-2.2.1-2-241909, FP7-HEALTH-2009-2.2.1-3-

242114, FP7-HEALTH-2013-2.2.1-2-603196, and FP7-HEALTH-2013-2.2.1-2-602478), Otsuka, Pfizer, Roche, Servier, Shire, Takeda, and Schering-Plough. Dr. Freitag has served as a consultant for Desitin regarding issues on ASD. Dr. Di Martino is a coauthor of the Italian version of the Social Responsiveness Scale, for which she may receive royalties. Her contribution is on behalf of the ABIDE and ABIDEII consortia. Dr. Rubia has received speaking honoraria from Eli Lilly, Medice, and Shire, and a grant from Shire for another project. Dr. Thompson received partial research support from Biogen, Inc. (Boston), for research unrelated to the topic of this manuscript. Dr. Buitelaar has served as a consultant, advisory board member, or speaker for Eli Lilly, Janssen-Cilag, Lundbeck, Medice, Novartis, Servier, Shire, and Roche, and he has received research support from Roche and Vifor. The remaining authors declare no competing interests.

## Additional information

Merel C. Postema [1], Daan van Rooij[2], Evdokia Anagnostou[3], Celso Arango[4], Guillaume Auzias [5], Marlene Behrmann[6], Geraldo Busatto Filho[7], Sara Calderoni[8,9], Rosa Calvo[10], Eileen Daly[11], Christine Deruelle[5], Adriana Di Martino[12], Ilan Dinstein [13], Fabio Luis S. Duran [7], Sarah Durston[14], Christine Ecker[15,16], Stefan Ehrlich[17], Damien Fair [18], Jennifer Fedor[19], Xin Feng[20], Jackie Fitzgerald[21,22], Dorothea L. Floris[2], Christine M. Freitag [15], Louise Gallagher[21,22], David C. Glahn[23,24], Ilaria Gori[25], Shlomi Haar [26], Liesbeth Hoekstra[2,27], Neda Jahanshad [16], Maria Jalbrzikowski[21], Joost Janssen[4], Joseph A. King [17], Xiang Zhen Kong[1], Luisa Lazaro[10], Jason P. Lerch[28,29], Beatriz Luna[19], Mauricio M. Martinho[30], Jane McGrath[21,22], Sarah E. Medland[31], Filippo Muratori[8,9], Clodagh M. Murphy [11,32], Declan G.M. Murphy [32,33], Kirsten O'Hearn[19], Bob Oranje[14], Mara Parellada[4], Olga Puig[10], Alessandra Retico [25], Pedro Rosa[7], Katya Rubia[34], Devon Shook[14], Margot J. Taylor[35], Michela Tosetti [8], Gregory L. Wallace[36], Fengfeng Zhou [20], Paul M. Thompson[16], Simon E. Fisher [1,37], Jan K. Buitelaar [2] & Clyde Francks[1,37]*

[1]Department of Language & Genetics, Max Planck Institute for Psycholinguistics, Nijmegen, the Netherlands. [2]Department of Cognitive Neuroscience, Donders Institute for Brain, Cognition and Behaviour, Donders Centre for Cognitive Neuroimaging, Radboud University Medical Centre, Nijmegen, The Netherlands. [3]Bloorview Research Institute, Holland Bloorview Kids Rehabilitation Hospital and Department of Pediatrics, University of Toronto, Toronto, USA. [4]Child and Adolescent Psychiatry Department, Gregorio Marañón General University Hospital, School of Medicine, Universidad Complutense, IiSGM, CIBERSAM, Madrid, Spain. [5]Institut de Neurosciences de la Timone, UMR 7289, Aix Marseille Université, CNRS, Marseille, France. [6]Department of Psychology, Carnegie Mellon University, Pittsburgh, PA, USA. [7]Laboratory of Psychiatric Neuroimaging (LIM-21), Departamento e Instituto de Psiquiatria, Hospital das Clinicas HCFMUSP, Faculdade de Medicina, Universidade de Sao Paulo, Sao Paulo, SP, Brazil. [8]IRCCS Stella Maris Foundation, viale del Tirreno 331, 56128 Pisa, Italy. [9]Department of Clinical and Experimental Medicine, University of Pisa, Pisa, Italy. [10]Department of Child and Adolescent Psychiatry and Psychology Hospital Clinic, Psychiatry Unit, Department of Medicine, 2017SGR881, University of Barcelona, IDIBAPS, CIBERSAM, Barcelona, Spain. [11]Department of Forensic and Neurodevelopmental Sciences, Institute of Psychiatry, Psychology & Neuroscience King's College London, London, UK. [12]Institute for Pediatric Neuroscience, NYU Child Study Center, New York, NY, USA. [13]Department of Psychology, Ben-Gurion University of the Negev, Beer Sheva, Israel. [14]Brain Center Rudolf Magnus, Department of Psychiatry, University Medical Center Utrecht, Utrecht, The Netherlands. [15]Department of Child and Adolescent Psychiatry, Psychosomatics and Psychotherapy, University Hospital, Goethe University Frankfurt am Main, Frankfurt, Germany. [16]Imaging Genetics Center, Mark and Mary Stevens Neuroimaging and Informatics Institute, Keck School of Medicine of USC, University of Southern California, Marina del Rey, CA, USA. [17]Department of Child and Adolescent Psychiatry, Division of Psychological and Social Medicine and Developmental Neurosciences, Faculty of Medicine, TU Dresden, Dresden, Germany. [18]Department of Behavioral Neuroscience, Oregon Health & Science University, Portland, OR, USA. [19]Department of Psychiatry, University of Pittsburgh, Pittsburgh, PA, USA. [20]BioKnow Health Informatics Lab, College of Computer Science and Technology, and Key Laboratory of Symbolic Computation and Knowledge Engineering of Ministry of Education, Jilin University, Changchun, Jilin 130012, China. [21]Department of Psychiatry, School of Medicine, Trinity College, Dublin, Ireland. [22]The Trinity College Institute of Neuroscience, Trinity College, Dublin, Ireland. [23]Department of Psychiatry, Boston Children's Hospital and Harvard Medical School, Boston, MA 02115-5724, USA. [24]Olin Neuropsychiatric Research Center, Hartford, CT, USA. [25]National Institute for Nuclear Physics, Pisa Division, Largo B. Pontecorvo 3, 56124 Pisa, Italy. [26]Department of Bioengineering, Imperial College London, London, UK. [27]Karakter Child and Adolescent Psychiatry University Centre, Nijmegen, The Netherlands. [28]Neurosciences and Mental Health, The Hospital for Sick Children, Toronto, ON, Canada. [29]Department of Medical Biophysics, University of Toronto, Toronto, ON, Canada. [30]Department of Neuropsychiatry,

Universidade Federal de Santa Maria, Santa Maria, Brazil. [31]Psychiatric Genetics, QIMR Berghofer Medical Research Institute, Brisbane, QLD, Australia. [32]Behavioural Genetics Clinic, Adult Autism Service, Behavioural and Developmental Psychiatry Clinical Academic Group, South London and Maudsley Foundation NHS Trust, London, UK. [33]The Sackler Institute for Translational Neurodevelopment, Institute of Psychiatry, Psychology & Neuroscience, King's College London, London, UK. [34]Institute of Psychiatry, Psychology and Neuroscience, Kings College London, London, UK. [35]Diagnostic Imaging Research, The Hospital for Sick Children, University of Toronto, Toronto, ON, Canada. [36]Department of Speech, Language, and Hearing Sciences, The George Washington University, Washington, DC, USA. [37]Donders Institute for Brain, Cognition and Behaviour, Radboud University, Nijmegen, The Netherlands. *email: clyde.francks@mpi.nl

