## [Peer Review File · Nature Communications]

Reviewers' comments:

Reviewer #1 (Remarks to the Author):

In this study Postema et al. used a large multisite sample of MR images from people with and without ASD to assess effects of cortical asymmetry. Data from ~1800 individuals aged 2-65 with and without ASD were assembled for the analyses. FreeSurfer was used to obtain cortical thickness and surface area measures from anatomical regions in the Desikan atlas and entered into linear mixed models that accounted for site-related clustering.

Significant asymmetry differences in ASD relative to the control sample were found for thickness overall and for several regions for thickness, with asymmetry generally reduced in ASD, and only one region for surface area. In subcortical regions examined, only the putamen showed a significant difference in volume asymmetry.

Importantly the authors report effect sizes which were relatively small, and demonstrate that they have sufficient power to detect effects in this range but that most previous work in this area has been underpowered.

This is an interesting and very clearly written paper. While the work is important, the authors acknowledge that the clinical utility of the findings may be limited given the small effect sizes. Nonetheless doing this work to establish that benchmark is important and commendable. I have relatively minor comments towards an improved version.

- 1) Some citations for the paragraph at the top of page 6 are needed to support claims of e.g. larger average volume
- 2) Line 69 – would add 'potentially' before 'due to'
- 3) Given the large sample size, I was curious why age- and sex- specific effects or interactions were not modeled, but only followed up posthoc in areas showing a diagnostic effect, particularly given the wide age range?
- 4) Page 12 line 217 – confusing in the same section to use n1 and n2 twice to mean different things
- 5) In the age-specific effects follow up I was confused as to why age was coded as a binary variable? More generally I wonder about the utility of including the full age range given the sparse sampling particularly at older ages?
- 6) Figure 1: color bar should be labeled and the *s are a bit clumsy – it's hard to actually see what they are meant to refer to on the surface.
- 7) Line 357: why a 'putative' effect?
- 8) Age and IQ were examined separately – perhaps a missed opportunity to understand the effect in the rostral ACC?
- 9) Given the small effect size reported, I wondered about difference in sensitivity to small cortical features in the 1.5T vs. 3T acquired data. Maybe worth running analyses on the subset of 3T acquired data?

Reviewer #2 (Remarks to the Author):

This is another interesting and important study from the ENIGMA consortium. The main strength of the paper is the large sample size and the use of a known and standardised pipeline. While the question is not particularly novel, the sample size allows a more definitive answer on the presence of brain asymmetry in autism.

There are some major issues to consider with the methodological approach used here.

The authors have removed outliers at various stages in their analyses. Complete removal of outliers is, in my experience, not typically encouraged by statisticians and where it does happen it would be more typical to present the outlier-removed results as a secondary/sensitivity analysis.

It is not clear what the authors did with regards to outlier removal and why. They wrote: "outliers were defined per group (cases/controls) and per dataset as those values above and below 1.5 times the interquartile range, and were excluded". It would be helpful if they clarified:

- Why this approach was necessary in more details. Why are outliers particularly a concern for this type of analysis? Did the authors consider less extreme treatment of outlier such as Winsorisation? The authors state how many datapoints were removed, but they do not give the important context of what fraction of the observations this was and whether this varies by site/caseness.
- What did they actually do? In this context the interquartile range must be taken relative to something (e.g. mean, median, or upper/lower quartiles). The latter option would be Tukey's fences method (basis for whisker end points in a Tukey boxplot), this method is common, but not universal. This has implications for evaluating the technique. If it is Tukey's method $\sim 0.7\%$ of normal data would be excluded erroneously. If it is the sample median $\pm 1.5\text{IQR}$ then $\sim 4\%$ of perfectly normally distributed data would be excluded. Because the authors repeat the process 3 times (L, R and LI) the expected rate of outliers in perfectly normal is $1 - (1 - \text{Rate})^3$ i.e. $\sim 2\%$ and $\sim 11\%$ respectively.
- Was outlier detection calculated for each combination of group and dataset, or once looking for outliers within each group (pooling datasets) and once looking for outliers within each dataset (pooling groups)?

There are also some concerns about the appropriateness of the technique:

- The effectiveness of this method (sensitivity to outliers) varies systematically with sample size. Because outliers are considered within each site and without reference to the final model this adds a difficult to quantify bias to the results.
- The outlier detection is conducted within site and caseness, but age and eTIV are not considered – some of the subjects are as young as 2 years old, presumably they will be outliers for most of the assessed measures? Again the effectiveness of the outlier identification will vary systematically with key covariates.
- The outlier detection is run repeatedly for each measure, a multivariate approach considering both L and R data could be run just once per measure and would capture unusual LI. Something like a bivariate confidence ellipse or Mahalanobis D2.

I would suggest reporting the results without excluding outliers as the primary analysis, but an alternative would be to use an outlier robust model such as the `robustlmm` package which fits huberised linear mixed effect models for potentially contaminated data. These models down-weight the influence of outliers in a data-driven manner.

Other minor issues:

The authors did not adjust for IQ in their analysis, and carried out a post hoc analysis in the patients only, providing a good justification for this. However, it would be helpful if they also performed the same post hoc analysis in the healthy controls, to explore the relationship between IQ and brain asymmetries in non-pathological conditions.

Terms such as "autistic individuals" should be avoided and replaced by "individuals with autism".

In the Introduction, it is unclear why only prevalence in the USA is reported.

Reviewer #3 (Remarks to the Author):

MS: NCOMMS-19-09029: Altered structural brain asymmetry in autism spectrum disorder: large-scale analysis via the ENIGMA consortium

General Comments:

This large-scale brain structure collaborative study investigated the structural brain asymmetry among 1774 subjects with autism spectrum disorders (ASD) and 1809 controls recruited from 54 sites (54 datasets) in terms of cortical thickness, surface area, and subcortical volume using Freesurfer analysis. The authors used "asymmetry indexes (AI), a widely used index in brain asymmetry studies, to estimate the structural brain asymmetry. The authors identified significant group (ASD-control) differences in 11 structural AIs: the total hemispheric average thickness AI; eight regional cortical thickness AIs including frontal regions (superior frontal, rostral middle frontal, medial orbitofrontal), temporal regions (fusiform, inferior temporal, superior temporal), and cingulate regions (rostral anterior, isthmus cingulate); one cortical regional surface area AI (the medial orbitofrontal cortex); and one subcortical volume AI (the putamen). These results survived multiple testing correction. The absolute value of the effect sizes presented by Cohen's d of these 11 significant results ranging from 0.09 to 0.16 is very low. In addition, there was significant sex by diagnosis interaction for the rostral anterior cingulate thickness but none sex by diagnosis interaction for any of the above 11 AI main findings. Moreover, within the ASD group, IQ was positively

The strengths of this work is the largest scale brain structural study examining the structural brain asymmetry with combining data from 54 sites using the same analysis protocol with sophisticated statistical analysis considering the random effect from the same site (lack of independence of observation within the same site) and multiple testing correction in the group differences analysis. Nevertheless, I have several major and minor comments as follows.

Specific Comments:

Abstract

1. The eight regional cortical thickness AIs included the major regions related to Default-Mode Network (DMN) in the frontal regions, temporal regions, and cingulate regions. This can be included in the abstract section to make a point of altered AI in DMN in ASD.

Introduction

1. In addition to the prevalence of ASD in the US, as ENIGMA consortium collected data from many countries, the ASD prevalence rates in the European countries, as well as other countries, are recommended to be included.

2. The authors reviewed the group differences in asymmetry in imaging studies regarding resting-state fMRI, DSI, and structural MRI. As sex- and age-by diagnosis interactions would be tested in this study and the influence of IQ and autistic severity on the AI would be investigated, too, a concise review about the age, sex, IQ and ASD symptom severity is suggested.

Methods

1. The sample:

i. Diagnosis: how many subjects were diagnosed as ASD according to the DSM-IV diagnostic criteria? Among these ASD subjects, the number of subjects in each of the diagnostic subgroups (Autistic Disorder, Asperger's Syndrome, Childhood Disintegrative Disorder, and Pervasive Developmental Disorder – Not Otherwise Specified (PDD-NOS). I am interested in knowing the number of subjects with Childhood Disintegrative Disorder. What instruments were used to make a diagnosis of ASD? Clinical diagnosis? ADI-R? ADOS? ASD diagnosis based on clinical assessment by board-certified psychiatrists or other recognized clinicians is acceptable. I am interested in knowing how to ensure that the controls did not have a diagnosis of ASD.

ii. Age range: In such a big dataset with wide-ranging age distribution is anticipated. Although the subjects with ages of 1.5 years were excluded from the analysis (how many subjects with such a young age?), ages of 2 to 4 were still included in the FreeSurfer analysis. Brain structure analysis of preschoolers (ages of 2-6) using FreeSurfer are very challenging and may create invalid data. Since this work has the largest dataset, the inclusion of ASD and control subjects whose ages of 6 and older is suggested.

iii. IQ: The IQ range is wide for both ASD and control groups. How many are under 70 for the ASD and control groups? For controls with IQ under 70, any diagnosis of developmental disorders was assessed? ,

iv. Comorbidity: Only one-third of ASD subjects were assessed for comorbid conditions. Among them, less than 10% showed at least one comorbid condition (ADHD, OCD, depression, anxiety,

and/or TS). The rate is quite low as compared to clinical sample even epidemiological samples. I wonder whether ADHD and the other psychiatric disorders were excluded in the recruitment procedure. ADHD is quite common in individuals with ASD, and it is hard to add ADHD as one of the exclusion criteria for ASD imaging studies.

What about psychiatric comorbid conditions in the controls?

2. Structural MRI scans, data acquisition, and analysis:

i. The data combined 54 images datasets acquired using different field strengths (1.5T or 3 T), and different cross-site scanners (different sequences and parameters). I read their protocols and do not find how the research team dealt with different scanners and field strengths. This is a basic issue that needs to be solved before data analysis of such large-scale data from 54 sites.

ii. The authors can conduct some sensitivity analyses by randomly taking away some datasets and running the analyses to see whether the results are similar.

iii. Regarding the quality control, more details of the procedures and criteria taken to deal with quality control are needed. Did they take care of motion issue? Please refer to the references, e.g., <https://www.ncbi.nlm.nih.gov/pubmed/23707591>

Renormalization of the imaging data before conducting FreeSurfer is needed rather than combining the output raw data of FreeSurfer from each site. Please refer to the reference, e.g.,

<https://ieeexplore.ieee.org/abstract/document/4141193/metrics#metrics>

<https://www.sciencedirect.com/science/article/pii/S1053811907007604>

iv. Page 10, the definition of outliers as the values above and below 1.5 times the interquartile range. Any reference for using this criterion as outliers?

Since 54 datasets were included in the analysis, did the authors found which sites or scanners, or the processing were associated with significantly higher rates of outliers? The authors have conducted the analyses with and without removing outliers. Whether the results were the same, regardless of removing outliers or not?

v. Page 10, the definition of outliers as the values above and below 1.5 times the interquartile range. Any reference for using this criterion as outliers?

Since 54 datasets were included in the analysis, did the authors found which sites or scanners, or the processing were associated with significantly higher rates of outliers? The authors have conducted analyses with and without removing outliers. Whether the results were the same, regardless of removing outliers or not?

3. Statistical Analysis:

i. The authors provide detailed description of statistical models used in the data analyses. To address the lack of independence within the same site, the Linear Mixed Model was used in all the analyses. The detailed modeling equation may not be necessary if there is a limited word count. I think the readers should be familiar with the effect size presented by Cohen's d. The equation of Cohen's d may not be necessary. However, the interpretation of the magnitude of effect size should be included (small, medium, and large effect sizes as Cohen's d 0.2 to 0.5, 0.5 to 0.8, and ≥ 0.8 , respectively [Cohen, 1988]). Cohen J (1988), *Statistical power analysis for the behavioral sciences* (2nd ed.) Hillsdale, NJ: Lawrence Earlbaum Associates

ii. Why did the authors only correlate IQ with AI results in the ASD group?

If the authors want to test these correlations, the analyses should be conducted for both groups separately.

iii. For ADOS correlations with AI measures, did the authors check any significant difference in the ADOS scores across sites? How many subjects have ADOS assessments? If there were significant cross-site differences in ADOS scores, the R-package used for normalization is based on ranking may not be able to normalize the ADOS scores across so many sites. If there were no significant differences, the authors' efforts may be OK.

Results:

1. Table 1 and Figure 1 are clear to the readers.

2. Page 19: There were no statistical significant sex- and age-by diagnosis interactions except sex by diagnosis interaction for the rostral anterior cingulate thickness AI. I wonder the main effects of

diagnosis still maintained significant after adding the interaction terms in the models.

3. Page 20: The authors need to mention that the correlation of ADOS severity score is not corrected for multiple testing.

Discussion

1. The eight regional cortical thickness AIs included the major regions related to Default-Mode Network (DMN) in the frontal regions, temporal regions, and cingulate regions. The findings are important regarding the role of altered brain asymmetry in DMN in individuals with ASD. Review and Discussions on the DMN in ASD are suggested.

2. The effect sizes are very low. Why did the authors estimate the power using a very low effect size? Given such a large sample, very low effect sizes may still generate significant results with a small p-value.

3. Despite the strengths of largest sample size on this topic, the issue of heterogeneity of subjects with ASD (controls, too), MRI field strengths, scanners, parameters, and sequences need to manage carefully using state of the art methodology.

4. Page 22, 1st paragraph: The main results need to be discussed.

5. Page 22, 3rd paragraph: The finding regarding the fusiform cortex is contradictory to the previous study by Dougherty et al. (2016) needs the interpretation and discussion.

6. Page 23, Lines 442-443: At least a reference is needed for this sentence.

7. Page 23, Line 447: ...due to limited statistical power in the earlier studies, which may have resulted in false positive findings. It should read as "negative" finding.

8. Line 463: how to interpret "lower-performing males"? Please do not just repeat the result but interpretation and discussions of the findings are needed.

9. Line 464: the only finding regarding ADOS correlation may be just by chance because of the uncorrected p-value. General speaking the AIs results were not correlated with autistic symptoms.

10. Line 471, the comorbidities are low.

11. Line 475, Why the authors did not test the effect of the medication use?

12. For neuropsychological and brain imaging studies, IQ needs to be controlled in the models.

Reviewer #1 (Remarks to the Author):

In this study Postema et al. used a large multisite sample of MR images from people with and without ASD to assess effects of cortical asymmetry. Data from ~1800 individuals aged 2-65 with and without ASD were assembled for the analyses. FreeSurfer was used to obtain cortical thickness and surface area measures from anatomical regions in the Desikan atlas and entered into linear mixed models that accounted for site-related clustering. Significant asymmetry differences in ASD relative to the control sample were found for thickness overall and for several regions for thickness, with asymmetry generally reduced in ASD, and only one region for surface area. In subcortical regions examined, only the putamen showed a significant difference in volume asymmetry. Importantly the authors report effect sizes which were relatively small, and demonstrate that they have sufficient power to detect effects in this range but that most previous work in this area has been underpowered. This is an interesting and very clearly written paper. While the work is important, the authors acknowledge that the clinical utility of the findings may be limited given the small effect sizes. Nonetheless doing this work to establish that benchmark is important and commendable.

Authors: Thank you very much for these supportive comments.

I have relatively minor comments towards an improved version.

1) Some citations for the paragraph at the top of page 6 are needed to support claims of e.g. larger average volume

Authors: To support this we have again cited the reviews referred to in the previous sentence, and also added further references:

Lucibello, S., Verdolotti, T., Giordano, F. M., Lapenta, L., Infante, A., Piludu, F., . . . Battini, R. (2019). Brain morphometry of preschool age children affected by autism spectrum disorder: Correlation with clinical findings. *Clin Anat*, 32(1), 143-150. doi:10.1002/ca.23252

Courchesne, E., Pierce, K., Schumann, C.M., Redcay, E., Buckwalter, J.A., Kennedy, D.P., Morgan, J., 2007. Mapping early brain development in autism. *Neuron* 56, 399–413.

Courchesne, E., Karns, C.M., Davis, H.R., Ziccardi, R., Carper, R.A., Tigue, Z.D., Pierce, K., Moses, P., Chisum, H.J., Lord, C., Lincoln, A.J., Pizzo, S., Schreibman, L., Haas, R.H., Akshoomoff, N., Courchesne, R.Y., 2001. Unusual brain growth patterns in early life in patients with autistic disorder: an MRI study. *Neurology* 57, 245–254

Retico, A., Giuliano A., Tancredi R., Cosenza A., Apicella F., Narzisi A., Biagi L, Tosetti M , Muratori F, Calderoni S. 2016. “The effect of gender on the neuroanatomy of children with autism spectrum disorders: A support vector machine case-control study.” *Molecular Autism* 7(1): 5.

2) Line 69 – would add ‘potentially’ before ‘due to’

Authors: Done

3) Given the large sample size, I was curious why age- and sex- specific effects or interactions were not modeled, but only followed up posthoc in areas showing a diagnostic effect, particularly given the wide age range?

Authors: We have now added this to the paper as a secondary analysis for all AIs, pages 15 (Methods section), 20 (Results section), and Tables S9-S14 (interaction term and stratification results for all AIs). Briefly, the addition of diag:age interaction to the primary model did not show any significant interaction effects after FDR correction. There was one significant diag:sex interaction, for the rostral anterior cingulate thickness AI. This was already remarked on in the paper, since this region showed an effect of diagnosis in the primary analysis (and had therefore already been subject to post-hoc analysis).

4) Page 12 line 217 – confusing in the same section to use n1 and n2 twice to mean different things

Authors: We have changed the second ‘n1’ and ‘n2’ to ‘x1’ and ‘x2’.

5) In the age-specific effects follow up I was confused as to why age was coded as a binary variable? More generally I wonder about the utility of including the full age range given the sparse sampling particularly at older ages?

Authors: We have now coded age as a continuous variable in all models where it is included as a covariate, to be consistent with the primary analysis (page 15). We have also added a sensitivity analysis where we excluded any individuals aged over 40 years, and the pattern of significant results remains the same as the primary analysis (pages 14(methods), 18 (results), and Tables S5-S7).

6) Figure 1: color bar should be labeled and the *s are a bit clumsy – it’s hard to actually see what they are meant to refer to on the surface.

Authors: The color bars have now been labeled with ‘Cohen’s d’. We have removed the symbols and left the colour to indicate the strongest effects.

7) Line 357: why a ‘putative’ effect?

We removed ‘putative’.

8) Age and IQ were examined separately – perhaps a missed opportunity to understand the effect in the rostral ACC?

Authors: We have now noted (page 21) that none of the interaction terms age*IQ, sex*IQ or age*sex*IQ were significant (all $P > 0.05$) when included in the case-only analysis model for this regional AI.

9) Given the small effect size reported, I wondered about difference in sensitivity to small

cortical features in the 1.5T vs. 3T acquired data. Maybe worth running analyses on the subset of 3T acquired data?

Authors: The large majority of datasets (45 out of 54) were 3T acquired. We have now added a sensitivity analysis conducted only in the 3T datasets (pages 14, 18, results for all AIs in Tables S5-S7). There were slight changes in significance, such that two of the diagnosis effects from the primary analysis (i.e., inferior temporal and isthmus cingulate thickness AIs) were no longer significant after false discovery rate correction, but three other effects now became significant (i.e., superior temporal thickness AI, fusiform surface area AI, and caudate nucleus AI). However, as the sample drops from 1778 cases and 1829 controls in the primary analysis to 1467 cases and 1574 controls in the 3T-only analysis, then slight changes in significance levels are expected, and do not necessarily indicate systematic differences of 3T and 1.5T data.

Reviewer #2 (Remarks to the Author):

This is another interesting and important study from the ENIGMA consortium. The main strength of the paper is the large sample size and the use of a known and standardised pipeline. While the question is not particularly novel, the sample size allows a more definitive answer on the presence of brain asymmetry in autism.

There are some major issues to consider with the methodological approach used here.

The authors have removed outliers at various stages in their analyses. Complete removal of outliers is, in my experience, not typically encouraged by statisticians and where it does happen it would be more typical to present the outlier-removed results as a secondary/sensitivity analysis.

It is not clear what the authors did with regards to outlier removal and why. They wrote: “outliers were defined per group (cases/controls) and per dataset as those values above and below 1.5 times the interquartile range, and were excluded”. It would be helpful if they clarified:

- Why this approach was necessary in more details. Why are outliers particularly a concern for this type of analysis? Did the authors consider less extreme treatment of outlier such as Winsorisation? The authors state how many datapoints were removed, but they do not give the important context of what fraction of the observations this was and whether this varies by site/caseness.
- What did they actually do? In this context the interquartile range must be taken relative to something (e.g. mean, median, or upper/lower quartiles). The latter option would be Tukey’s fences method (basis for whisker end points in a Tukey boxplot), this method is common, but not universal. This has implications for evaluating the technique. If it is Tukey’s method $\sim 0.7\%$ of normal data would be excluded erroneously. If it is the sample median $\pm 1.5\text{IQR}$ then $\sim 4\%$ of perfectly normally distributed data would be excluded. Because the authors repeat the process 3 times (L, R and LI) the expected rate of outliers in perfectly normal is $1 - (1 - \text{Rate})^3$ i.e. $\sim 2\%$ and $\sim 11\%$ respectively.

- Was outlier detection calculated for each combination of group and dataset, or once looking for outliers within each group (pooling datasets) and once looking for outliers within each dataset (pooling groups)?

There are also some concerns about the appropriateness of the technique:

- The effectiveness of this method (sensitivity to outliers) varies systematically with sample size. Because outliers are considered within each site and without reference to the final model this adds a difficulty to quantify bias to the results.
- The outlier detection is conducted within site and caseness, but age and eTIV are not considered – some of the subjects are as young as 2 years old, presumably they will be outliers for most of the assessed measures? Again the effectiveness of the outlier identification will vary systematically with key covariates.
- The outlier detection is run repeatedly for each measure, a multivariate approach considering both L and R data could be run just once per measure and would capture unusual LI. Something like a bivariate confidence ellipse or Mahalanobis D2.

I would suggest reporting the results without excluding outliers as the primary analysis, but an alternative would be to use an outlier robust model such as the robustlmm package which fits huberised linear mixed effect models for potentially contaminated data. These models down-weight the influence of outliers in a data-driven manner.

Authors: Many thanks to the reviewer for this advice. We have now handled outliers as recommended by the reviewer, i.e. no outlier exclusion for the primary analysis, and the asymmetry indexes winsorised for a sensitivity analysis. We used a threshold of $k=3$ for winsorization (i.e., so that the two most upper/lower values were brought back to the value of the 3rd upper/lower value) for a confirmatory analysis. This threshold was chosen after visual inspection of outlier patterns.

Therefore no datapoints have been excluded in the revised version of the paper. The results remain largely as before - see changes to pages 16,17 (results) and the results tables (Table 1, Tables S2-S4). In addition, the significant effects from the new, primary analysis are the same ones that are significant in the winsorised analysis, except that the effect of diagnosis in medial orbitofrontal surface area AI did not survive multiple testing correction in the winsorized analysis (although the P value barely changed) (Table S6). Note that the asymmetry index $(L-R)/(L+R)$ does not necessarily scale with L and R due to its denominator (now noted on page 11), so that larger or smaller brains are not necessarily more likely to be outliers for this asymmetry index. In any case, we have also added a sensitivity analysis excluding the subjects aged less than 6 years (in response to a comment from reviewer 3), and the results are largely unchanged by this (see below in response to reviewer 3, and Tables S5-S7). Additionally, excluding individuals >40years old did not change the pattern of results (see response to reviewer 1 above).

Other minor issues:

The authors did not adjust for IQ in their analysis, and carried out a post hoc analysis in the patients only, providing a good justification for this. However, it would be helpful if they also performed the same post hoc analysis in the healthy controls, to explore the relationship between IQ and brain asymmetries in non-pathological conditions.

Authors: We have now added this post hoc analysis in controls too (pages 15, 21 and Table S15). In the patients, only the rostral ACC thickness asymmetry shows an association with IQ, whereas in the controls this asymmetry shows no association with IQ, but a different asymmetry shows an association with IQ (superiorfrontal thickness). This perhaps underlines the uncertain nature of effects arising from these exploratory, secondary analyses of the data, where FDR correction was not applied. Alternatively, brain-IQ associations may be different in cases and controls. Both possibilities are now acknowledged in the paper (Discussion, page 26).

Terms such as “autistic individuals” should be avoided and replaced by “individuals with autism”.

Authors: We have changed to ‘individuals with autism’.

In the Introduction, it is unclear why only prevalence in the USA is reported.

Authors:

Authors: We have now cited a reference on global prevalence in the first paragraph:

Elsabbagh, M. et al. Global prevalence of autism and other pervasive developmental disorders. *Autism research : official journal of the International Society for Autism Research* 5, 160-179, doi:10.1002/aur.239 (2012).

Reviewer #3 (Remarks to the Author):

MS: NCOMMS-19-09029: Altered structural brain asymmetry in autism spectrum disorder: large-scale analysis via the ENIGMA consortium

General Comments:

This large-scale brain structure collaborative study investigated the structural brain asymmetry among 1774 subjects with autism spectrum disorders (ASD) and 1809 controls recruited from 54 sites (54 datasets) in terms of cortical thickness, surface area, and subcortical volume using Freesurfer analysis. The authors used “asymmetry indexes (AI), a widely used index in brain asymmetry studies, to estimate the structural brain asymmetry. The authors identified significant group (ASD-control) differences in 11 structural AIs: the total hemispheric average thickness AI; eight regional cortical thickness AIs including frontal regions (superior frontal, rostral middle frontal, medial orbitofrontal), temporal regions (fusiform, inferior temporal, superior temporal), and cingulate regions (rostral anterior, isthmus cingulate); one cortical regional surface area AI (the medial orbitofrontal cortex); and one subcortical volume AI (the putamen). These results survived multiple testing correction. The absolute value of the effect sizes presented by Cohen’s *d* of these 11 significant results ranging from 0.09 to 0.16 is very low. In addition, there was significant sex by diagnosis interaction for the rostral anterior cingulate thickness but none sex by diagnosis interaction for any of the above 11 AI main findings. Moreover, within the ASD group, IQ was positively

The strengths of this work is the largest scale brain structural study examining the structural brain asymmetry with combining data from 54 sites using the same analysis protocol with

sophisticated statistical analysis considering the random effect from the same site (lack of independence of observation within the same site) and multiple testing correction in the group differences analysis. Nevertheless, I have several major and minor comments as follows.

Specific Comments:

Abstract

1. The eight regional cortical thickness AIs included the major regions related to Default-Mode Network (DMN) in the frontal regions, temporal regions, and cingulate regions. This can be included in the abstract section to make a point of altered AI in DMN in ASD.

Authors: We have noted a possible overlap with the DMN in the Discussion (see below where the reviewer also notes this point for the Discussion). However, since our data are structural and based on an atlas of pre-defined anatomical regions, we do not regard altered DMN asymmetry as a primary conclusion from our study, and have not added it to the abstract.

Introduction

1. In addition to the prevalence of ASD in the US, as ENIGMA consortium collected data from many countries, the ASD prevalence rates in the European countries, as well as other countries, are recommended to be included.

Authors: We have now cited a reference on global prevalence in the first paragraph (see also comment reviewer 2):

Elsabbagh, M. et al. Global prevalence of autism and other pervasive developmental disorders. *Autism research : official journal of the International Society for Autism Research* 5, 160-179, doi:10.1002/aur.239 (2012).

2. The authors reviewed the group differences in asymmetry in imaging studies regarding resting-state fMRI, DSI, and structural MRI. As sex- and age-by diagnosis interactions would be tested in this study and the influence of IQ and autistic severity on the AI would be investigated, too, a concise review about the age, sex, IQ and ASD symptom severity is suggested.

Authors: We have now briefly mentioned some further information on these aspects in the last paragraph of the introduction, while being mindful of the overall length of the article (page 8). For this purpose, the following articles were cited (the last of these goes into some detail on these issues, and is very recent):

Kong, X. Z. et al. Mapping cortical brain asymmetry in 17,141 healthy individuals worldwide via the ENIGMA Consortium. *Proc Natl Acad Sci U S A*, doi:10.1073/pnas.1718418115 (2018).

Guadalupe, T. et al. Human subcortical brain asymmetries in 15,847 people worldwide reveal effects of age and sex. *Brain Imaging Behav*, doi:10.1007/s11682-016-9629-z (2016).

Bedford, S. A. et al. Large-scale analyses of the relationship between sex, age and intelligence quotient heterogeneity and cortical morphometry in autism spectrum disorder. *Mol Psychiatry*, doi:10.1038/s41380-019-0420-6 (2019).

Methods

1. The sample:

i. Diagnosis: how many subjects were diagnosed as ASD according to the DSM-IV diagnostic criteria? Among these ASD subjects, the number of subjects in each of the diagnostic subgroups (Autistic Disorder, Asperger's Syndrome, Childhood Disintegrative Disorder, and Pervasive Developmental Disorder – Not Otherwise Specified (PDD-NOS). I am interested in knowing the number of subjects with Childhood Disintegrative Disorder. What instruments were used to make a diagnosis of ASD? Clinical diagnosis? ADI-R? ADOS? ASD diagnosis based on clinical assessment by board-certified psychiatrists or other recognized clinicians is acceptable. I am interested in knowing how to ensure that the controls did not have a diagnosis of ASD.

Authors: All datasets were collected during the period when DSM-IV and DSM-IV-TR were the common classification systems, between 1994 and 2013, and the clinical diagnosis of ASD was made according to the DSM-IV criteria. The datasets were collected in a variety of different countries, and intended originally as separate studies. Nonetheless, all subjects were diagnosed based on clinical diagnosis by a clinically experienced and board certified psychiatrist/psychologist. This was a criterion for admission of a dataset into the ENIGMA-ASD cohort. We have now clarified these aspects in the manuscript (page 9). Most experts agree that clinical diagnosis is still the gold standard, and that additional confirmation of the clinical diagnosis by research instruments as ADOS and ADI-R is fine and relevant, but these instruments do not map perfectly on the clinical diagnosis.

In the DSM-5 scheme, the DSM-IV subtypes of ASD were dropped. Instead, one overall category of Autism Spectrum Disorders (ASD) was chosen. This was done for good reasons. The subtyping of ASD proved to be very unreliable across sites, since even top expert centers adhered to different diagnostic algorithms. Reliability however was very good to excellent at the level of the overall ASD category. Also, previous data are very inconsistent as to specific neurobiological underpinnings of the older clinical subtypes of ASD. For these reasons, we have not collated information on DSM-IV subtyping across the datasets. We have now explained this (page 9) and cited the following paper:

<https://www.annualreviews.org/doi/full/10.1146/annurev-clinpsy-032814-112745>

However, all of our datasets are in principle focused on Autistic Disorder, not any of the older subgroups.

ADOS was available for 878 of the cases, which permitted severity score analysis in relation to brain asymmetry, and this information and analysis are included in the manuscript (see further below). We do not have data from other instruments in sufficient numbers to contribute to the present study.

Inevitably there was not a homogenous assessment/recruitment process for controls across these many legacy datasets, but the overwhelming majority (see point 1.iii below) were typically developing/healthy, and no controls showed features that might have met a diagnosis of ASD.

All of these aspects have now been made explicit in the Methods (pages 9,10).

ii. Age range: In such a big dataset with wide-ranging age distribution is anticipated. Although the subjects with ages of 1.5 years were excluded from the analysis (how many subjects with such a young age?), ages of 2 to 4 were still included in the FreeSurfer analysis. Brain structure analysis of preschoolers (ages of 2-6) using FreeSurfer are very challenging and may create invalid data. Since this work has the largest dataset, the inclusion of ASD and control subjects whose ages of 6 and older is suggested.

Authors: In our primary analysis we have retained all subjects, but we have added a new sensitivity analysis where we confirm that the results are largely unchanged after removing all subjects aged below 6 years (113 cases and 64 controls were removed for this)(pages 14, and Tables S5-S7). Briefly, conducting our analysis on individuals aged 6 years and older, as compared to total sample, one new association now surpassed the multiple testing correction threshold (fusiform surface area AI), and one previously significant association dropped below the multiple testing correction threshold (isthmus cingulate thickness AI), but again the actual P values changed only slightly. These subtle changes do not necessarily indicate that exclusion of younger ages improved signal to noise in the data. We have also cited a study (page 14) which found that FreeSurfer segmentations in preschoolers were generally of good quality even before visual QC (while all data used for the current study went through the ENIGMA visual QC protocol): Gori et al, J Neuroimaging 2015, <https://www.ncbi.nlm.nih.gov/pubmed/26214066>

iii. IQ: The IQ range is wide for both ASD and control groups. How many are under 70 for the ASD and control groups? For controls with IQ under 70, any diagnosis of developmental disorders was assessed? ,

Authors: We have now stated in the paper (pages 9,10) that there were 85 subjects with IQ below 70, of which 19 were controls. In these subjects the exclusion of ASD diagnosis was performed by a senior child psychiatrist. Eighteen of these controls came from the FSM dataset and were diagnosed with ID.

iv. Comorbidity: Only one-third of ASD subjects were assessed for comorbid conditions. Among them, less than 10% showed at least one comorbid condition (ADHD, OCD, depression, anxiety, and/or TS). The rate is quite low as compared to clinical sample even epidemiological samples. I wonder whether ADHD and the other psychiatric disorders were excluded in the recruitment procedure. ADHD is quite common in individuals with ASD, and it is hard to add ADHD as one of the exclusion criteria for ASD imaging studies. What about psychiatric comorbid conditions in the controls?

Authors: We now acknowledge the relative lack of information on comorbid conditions as a limitation in the Discussion (page 26). As mentioned above, the 54 datasets were collected over the last 25 years in many countries, as separate studies, according to various recruitment and selection criteria. For example, until DSM-V came into use, a comorbid diagnosis of ADHD and ASD was not even possible. Nonetheless we expect that the 54 datasets capture the research-clinical ASD population well, even if comorbidities were not recorded in many of them.

There was not a homogenous assessment/recruitment process for controls across these datasets, but the overwhelming majority (see point 1.iii above) were typically developing/healthy at the time of MRI, and no controls showed features that might have met a diagnosis of ASD (now made clear in the Methods).

2. Structural MRI scans, data acquisition, and analysis:

i. The data combined 54 images datasets acquired using different field strengths (1.5T or 3 T), and different cross-site scanners (different sequences and parameters). I read their protocols and do not find how the research team dealt with different scanners and field strengths. This is a basic issue that needs to be solved before data analysis of such large-scale data from 54 sites.

ii. The authors can conduct some sensitivity analyses by randomly taking away some datasets and running the analyses to see whether the results are similar.

Authors: We indeed found in previous meta-analysis of 99 healthy control or general population datasets (Kong et al., PNAS 2018) that there was heterogeneity between datasets for the same regional asymmetry indexes as used in the present study. Accordingly, for our analyses in the present study of ASD, 'Dataset' was included as a random variable (i.e. each dataset received a random intercept, thus levelling all datasets when modelling the effects of diagnosis and other covariates) to account for differences between datasets, which can include demographic, clinical and technical heterogeneity. We have made clearer in the revised version that the analysis was based on a 'mixed-effects random-intercept model' (page 11).

The primary purpose of our study, based on 54 datasets that were originally collected as separate studies, was to assess the total combined evidence for effects over all of the available datasets, while explicitly allowing for heterogeneity between datasets through the use of random intercepts, and finally adding sensitivity and secondary analyses with respect to relevant variables. We now include five different sensitivity analyses: in 3T-only data (see response to reviewer 1, above), removing the youngest subjects (see further above), removing the oldest subjects (above), including a non-linear age adjustment (this was already in the paper), and analysis with winsorization of outlier datapoints (see above), as well as secondary analyses of age:diagnosis interactions, sex:diagnosis interactions, IQ, ADOS scores, and medication.

The heterogeneity mentioned by the reviewer is a feature of the current field. That is, as long as researchers publish many separate papers based on single datasets collected in particular ways, the field overall has the same problem. It can be considered a strength of our study that our 54 datasets capture this real-world heterogeneity, while the results represent the total combined evidence in these datasets. We have expanded on these points in the Discussion (page 28).'

iii. Regarding the quality control, more details of the procedures and criteria taken to deal with quality control are needed. Did they take care of motion issue? Please refer to the references, e.g., <https://www.ncbi.nlm.nih.gov/pubmed/23707591>
Renormalization of the imaging data before conducting FreeSurfer is needed rather than combining the output raw data of FreeSurfer from each site. Please refer to the reference, e.g.,

<https://ieeexplore.ieee.org/abstract/document/4141193/metrics#metrics>

<https://www.sciencedirect.com/science/article/pii/S1053811907007604>

Authors: We have expanded our description of the ENIGMA image QC protocol (page 10-11), which now includes information on visual inspection (briefly, checking a series of internal slices for all subjects, and 3D visualization for any extreme values; the url for the full protocol is also cited).

Regarding head motion, the reference that the reviewer suggested is mainly on motion issues in functional MRI data, which involves time series data. Motion can also impact anatomical MRI data, but we are not aware of an approach to automatically detect or control this. If motion might have caused a deterioration of Freesurfer performance in some cases, then the ENIGMA visual QC protocol was designed to detect such errors. Regarding renormalization, this is incorporated into the FreeSurfer pipeline that we have used, i.e., recon-all, and we have noted the point in the revised methods section (page 10). From the Freesurfer users email forum:

A user asked: "I read this article "Atlas Renormalization for Improved Brain MR Image Segmentation Across Scanner Platforms(2007)" and I wonder if these steps "normalization" and "normalization2" do the Atlas Renormalization which mentioned at that article or should I use another command myself to do the Atlas Renormalization that the article proposed? (sorry if the question is basic)".

Bruce Fischl (leading developer of FreeSurfer) replied: "those steps are already incorporated into recon-all - you don't need to do anything special".

iv. Page 10, the definition of outliers as the values above and below 1.5 times the interquartile range. Any reference for using this criterion as outliers? Since 54 datasets were included in the analysis, did the authors find which sites or scanners, or the processing were associated with significantly higher rates of outliers? The authors have conducted the analyses with and without removing outliers. Whether the results were the same, regardless of removing outliers or not?

Authors: As outlined in our response to reviewer 2, we have now changed our approach to outliers. The primary analysis now includes all datapoints without any outlier adjustment or removal. A sensitivity analysis based on winsorization returns the same set of significant results, with the exception of the medial orbitofrontal surface are AI which changes significance slightly, so that it is no longer significant after FDR control.

v. Page 10, the definition of outliers as the values above and below 1.5 times the interquartile range. Any reference for using this criterion as outliers? Since 54 datasets were included in the analysis, did the authors find which sites or scanners, or the processing were associated with significantly higher rates of outliers? The authors have conducted analyses with and without removing outliers. Whether the results were the same, regardless of removing outliers or not?

Authors: This paragraph was duplicated. Answer as above.

3. Statistical Analysis:

i. The authors provide detailed description of statistical models used in the data analyses. To address the lack of independence within the same site, the Linear Mixed Model was used in

all the analyses. The detailed modeling equation may not be necessary if there is a limited word count. I think the readers should be familiar with the effect size presented by Cohen's d . The equation of Cohen's d may not be necessary. However, the interpretation of the magnitude of effect size should be included (small, medium, and large effect sizes as Cohen's d 0.2 to 0.5, 0.5 to 0.8, and ≥ 0.8 , respectively [Cohen, 1988]). Cohen J (1988), Statistical power analysis for the behavioral sciences (2nd ed.) Hillsdale, NJ: Lawrence Earlbaum Associates

Authors: All effects in this study were $d < 0.2$, so it is not clear what terminology to adopt from this scheme. Regardless, the low effect sizes are made clear and interpreted/discussed throughout the paper, including the abstract.

ii. Why did the authors only correlate IQ with AI results in the ASD group?
If the authors want to test these correlations, the analyses should be conducted for both groups separately.

Authors: We have now included IQ analysis in the controls, as one of the secondary analyses. See response to Reviewer 2 (above) who made the same point.

iii. For ADOS correlations with AI measures, did the authors check any significant difference in the ADOS scores across sites? How many subjects have ADOS assessments? If there were significant cross-site differences in ADOS scores, the R-package used for normalization is based on ranking may not be able to normalize the ADOS scores across so many sites. If there were no significant differences, the authors' efforts may be OK.

Authors: 878 cases have ADOS information (stated on page 9). In the revised paper we take a simpler approach to adjusting this score, i.e. using \log_{10} transformation to reduce skewing (page 16, Table S16, Figure S1). Therefore this transformation was now done at the individual level, unbiased by dataset. The analysis model is then $AI = \log_{10}ADOS + age + sex + dataset(\text{random intercept})$, allowing for dataset heterogeneity.

Results:

1. Table 1 and Figure 1 are clear to the readers.
2. Page 19: There were no statistical significant sex- and age-by diagnosis interactions except sex by diagnosis interaction for the rostral anterior cingulate thickness AI. I wonder the main effects of diagnosis still maintained significant after adding the interaction terms in the models.

Authors: Tables S9-S14 show the significance of main effects after adding these interaction terms. Adding sex*diagnosis interaction terms only slightly changed the P values for the main effects of diagnosis. Adding age*diagnosis interaction terms largely reduced the significance of the main effects of diagnosis, even though the age*diagnosis terms themselves were not significant (all $P > 0.05$ for the AIs which were significant in the primary analysis). However, for the AIs with significant effects of diagnosis in the primary analysis, we note (page 21) that adding age*diagnosis terms also increased the AIC and BIC scores, indicating poorer model fit when including these non-significant age*diagnosis interaction terms. The AIC and BIC are now shown in Table S9-S14.

3. Page 20: The authors need to mention that the correlation of ADOS severity score is not corrected for multiple testing.

Authors: This is now mentioned (page 21). This is appropriate for secondary, exploratory analyses of significant effects which had arisen from the primary analysis (where multiple testing correction was applied).

Discussion

1. The eight regional cortical thickness AIs included the major regions related to Default-Mode Network (DMN) in the frontal regions, temporal regions, and cingulate regions. The findings are important regarding the role of altered brain asymmetry in DMN in individuals with ASD. Review and Discussions on the DMN in ASD are suggested.

Authors: We have now mentioned in the Discussion (page 24) that our affected regions correspond roughly to the DMN, with the caveat that our regions are defined structurally. We also cite a paper that has reported atypical DMN lateralisation in ASD (Nielsen et al <https://molecularautism.biomedcentral.com/articles/10.1186/2040-2392-5-8>), and two others that have reported bilateral DMN changes in ASD (Nunes, A. S., Peatfield, N., Vakorin, V. & Doesburg, S. M. Idiosyncratic organization of cortical networks in autism spectrum disorder. *Neuroimage* 190, 182-190, doi:10.1016/j.neuroimage.2018.01.022 (2019). Uddin, L. Q. The self in autism: an emerging view from neuroimaging. *Neurocase* 17, 201-208, doi:10.1080/13554794.2010.509320 (2011)). We also note that our results may support a role of altered lateralization of the DMN in ASD, warranting further investigations in this direction.

2. The effect sizes are very low. Why did the authors estimate the power using a very low effect size? Given such a large sample, very low effect sizes may still generate significant results with a small p-value.

Authors: We did not estimate power. Rather, we calculated the minimum effect size that would be needed to achieve 80% power in our study, given its overall sample size (page 13). This tells us, for example, that we may have missed real effects smaller than $d=0.13$.

3. Despite the strengths of largest sample size on this topic, the issue of heterogeneity of subjects with ASD (controls, too), MRI field strengths, scanners, parameters, and sequences need to be managed carefully using state of the art methodology.

Authors: See our answer further above, where the reviewer raised the same point about dataset heterogeneity. The primary purpose of a study such as ours, based on multiple datasets that were originally collected as separate studies, is to assess the total combined evidence for effects over all of the available datasets, while explicitly allowing for heterogeneity between datasets through the use of random intercepts (discussed on page 28). The heterogeneity of the datasets is representative of the broader field.

4. Page 22, 1st paragraph: The main results need to be discussed.

Authors: Here we have added a possible link to the DMN (see points above).

5. Page 22, 3rd paragraph: The finding regarding the fusiform cortex is contradictory to the previous study by Dougherty et al. (2016) needs the interpretation and discussion.

Authors: We have now noted (pages 24,25) that Dougherty *et al.* went on to show that their change in volume asymmetry was driven by a change in surface area asymmetry, not thickness asymmetry. In fact we see a nominally (unadjusted for multiple testing) significant change in surface area asymmetry for this region, in the same direction as reported by Dougherty. In general this supports the approach of studying thickness as distinct from surface area, as noted on page 25.

6. Page 23, Lines 442-443: At least a reference is needed for this sentence.

Authors: The sentence is about our interpretation of our own findings, not a reference to prior work.

7. Page 23, Line 447:...due to limited statistical power in the earlier studies, which may have resulted in false positive findings. It should read as “negative” finding.

Authors: The point here is not about negative findings. Low power also reduces the likelihood that a statistically significant result reflects a true effect, as discussed by Button et al. 2013 (cited in our manuscript). We have updated the wording at this point in the discussion to make it clearer.

8. Line 463: how to interpret “lower-performing males”? Please do not just repeat the result but interpretation and discussions of the findings are needed.

Authors: We regard this finding as tentative because it arose from our secondary, exploratory analyses, where we did not use multiple testing correction (as noted in the Discussion, page 26). In this case we note the result in plain language for the reader, but do not wish to speculate on mechanisms.

9. Line 464: the only finding regarding ADOS correlation may be just by chance because of the uncorrected p-value. General speaking the AIs results were not correlated with autistic symptoms.

Authors: Similarly to the point above, we have noted in the Discussion (page 26) the tentative nature of our findings from exploratory analysis, and also that most of these results were negative.

10. Line 471, the comorbidities are low.

Authors: See our answer further above, where the reviewer raised the same issue.

11. Line 475, Why the authors did not test the effect of the medication use?

Authors: We have added an analysis of medication use to the paper (pages 16, 22, Table S17). Only 214 of 832 cases with information on this aspect used some form of medication. None of the AIs with significant effects of diagnosis in the primary analysis showed significant effects of medication (all uncorrected $P > 0.05$).

12. For neuropsychological and brain imaging studies, IQ needs to be controlled in the models.

Authors: We investigated IQ post hoc, but did not control for it as a nuisance variable in the primary analysis. Reviewer 2 agreed with this approach, noting that ‘The authors did not adjust for IQ in their analysis, and carried out a post hoc analysis ... , providing a good justification for this.’ The reason is as explained in the Discussion (page 27): ‘... lower average IQ was clearly part of the ASD phenotype in our total combined dataset ... so that including IQ as a confounding factor in case-control analysis might have reduced the power to detect an association of diagnosis with asymmetry. This would occur if underlying susceptibility factors contribute both to altered asymmetry and reduced IQ, as part of the ASD phenotype.’

REVIEWERS' COMMENTS:

Reviewer #1 (Remarks to the Author):

I thank the authors for their thoughtful responses to my comments and have nothing further.

Reviewer #2 (Remarks to the Author):

I am satisfied with the revised version

Reviewer #3 (Remarks to the Author):

The authors have answered all the comments and revised the manuscript with additional secondary analyses and revised tables and new supplementary tables according to the comments. I have no further questions for this well-written paper.

Author responses to reviewer comments, manuscript NCOMMS-19-09029A

Reviewer #1 (Remarks to the Author):

I thank the authors for their thoughtful responses to my comments and have nothing further.

Reviewer #2 (Remarks to the Author):

I am satisfied with the revised version

Reviewer #3 (Remarks to the Author):

The authors have answered all the comments and revised the manuscript with additional secondary analyses and revised tables and new supplementary tables according to the comments.

I have no further questions for this well-written paper.

Authors: Many thanks to the reviewers for these supportive comments.